# Chemical composition of ultrafine aerosol particles in central Amazonia during the wet season

Hayley S. Glicker[1], Michael J. Lawler[1], John Ortega[1], Suzane S. de Sá[2], Scot T. Martin[2,3], Paulo Artaxo[4], Oscar Vega Bustillos[5], Rodrigo de Souza[6], Julio Tota[7], Annmarie Carlton[1], and James N. Smith[1*]

[1] Department of Chemistry, University of California, Irvine, CA 92697 USA
[2] School of Engineering and Applied Sciences, Harvard University, Cambridge, Massachusetts 02138 USA
[3] Department of Earth and Planetary Sciences, Harvard University, Cambridge, Massachusetts 02138 USA
[4] Institute of Physics, University of São Paulo, Rua do Matão 1371, 05508-090, São Paulo, Brazil
[5] Instituto de Pesquisas Energéticas e Nucleares, São Paulo, Brazil
[6] Universidade do Estado do Amazonas, Manaus, AM, Brazil
[7] Institute of Engineering and Geoscience, Federal University of West Pará, Santarém, PA, Brazil

*Correspondence to:* James N. Smith (jimsmith@uci.edu)

**Abstract** Central Amazonia serves as an ideal location to study atmospheric particle formation since it often represents nearly natural, pre-industrial conditions but can also experience periods of anthropogenic influence due to the presence of emissions from large metropolitan areas like Manaus, Brazil. Ultrafine (sub-100 nm diameter) particles are often observed in this region, although new particle formation events seldom occur near the ground despite being readily observed in other forested regions with similar emissions of volatile organic compounds. This study focuses on identifying the chemical composition of ultrafine particles as a means of determining the chemical species and mechanisms that may be responsible for new particle formation and growth in the region. These measurements were performed during the wet season as part of the GoAmazon2014/5 field campaign at a site located 70 km southwest of Manaus. A Thermal Desorption Chemical Ionization Mass Spectrometer (TDCIMS) characterized the most abundant compounds detected in ultrafine particles. Two time periods representing distinct influences on aerosol composition, which we label as "anthropogenic" and "background" periods, were studied as part of a larger ten-day period of analysis. Higher particle number concentrations were measured during the anthropogenic period, and modelled back-trajectory frequencies indicate transport of emissions from the Manaus metropolitan area. During the background period there were much lower number concentrations and back-trajectory frequencies showed that air masses arrived at the site predominantly from the forested regions to the north and northeast. TDCIMS-measured constituents also show distinct differences between the two observational periods. Although bisulfate was detected in particles throughout the ten-day period, the anthropogenic period had higher levels of particulate bisulfate overall. Ammonium and trimethyl ammonium were positively correlated with bisulfate. The background period had distinct diurnal patterns of particulate cyanate and acetate, while oxalate remained relatively constant during the ten-day period. 3-Methylfuran, a thermal decomposition product of particulate phase isoprene epoxydiol (IEPOX), was the dominant species measured in the positive ion mode. Principal Component Analysis (PCA) was performed on the TDCIMS-measured ion abundance and Aerosol Mass Spectrometer (AMS) mass concentration data. Two different hierarchical clusters representing unique influences arise: one comprising ultrafine particulate acetate, hydrogen oxalate, cyanate, trimethyl ammonium and 3-methylfuran and another made up of ultrafine particulate bisulfate, chloride, ammonium

and potassium. A third cluster separated AMS-measured species from the two TDCIMS-derived clusters, indicating
different sources or processes in ultrafine aerosol particle formation compared to larger submicron-sized particles.

**1. Introduction**

Atmospheric aerosols are ubiquitous in the troposphere and, organics contribute a large fraction to their chemical
composition (Jimenez et al., 2009). Models continue to have difficulty estimating the organic contribution to aerosols
in regions with both biogenic and anthropogenic influence (Shrivastava et al., 2017). Anthropogenic emissions have
increased with global population and the resulting influences of such emissions on secondary organic aerosol (SOA)
formation continue to be assessed (Hofmann, 2015). The reactive chemistry of organics in the presence of different
regulating species from urban sources, like sulfur dioxide ($SO_2$) and oxides of nitrogen, remains uncertain (Shrivastava
et al., 2017), although recent efforts have successfully incorporated this chemistry into air quality models simulated
for the southeastern United States (Carlton et al., 2018). Models are unable to predict the relationships between particle
physico-chemical properties and cloud formation and precipitation (IPCC, 2013). Reducing this uncertainty requires
an understanding of the mechanisms by which particles form and grow in the atmosphere, which mostly determine
the potential of these particles to serve as cloud condensation nuclei (CCN).
The Amazon basin is an ideal location to study how biogenic emissions, anthropogenic trace gases and oxidants, and
biomass burning impact the number and composition of atmospheric aerosol particles. The Amazon basin is one of
the few remaining tropical regions on Earth in which near-natural conditions, free of direct anthropogenic influence,
can be found. It has been referred to as the "Green Ocean," since particle concentrations can be as low as that seen
over the ocean and, like the marine atmosphere, small changes in particle properties can have a major impact on clouds
and climate (Andreae et al., 2004). While isoprene is the most abundantly emitted biogenic volatile organic compound
(BVOC), monoterpenes and sesquiterpenes are observed in amounts potentially sufficient to influence particle
composition (Alves et al., 2016; Jardine et al., 2015, 2011; Yáñez-Serrano et al., 2015; Yee et al., 2018). While, on an
annual basis, aerosol particle sources in the Amazon basin are dominated by the oxidation of BVOCs by OH and $O_3$,
in many parts of the Amazon, anthropogenic emissions of trace gases and oxidants, as well as human-caused-biomass
burning, can have a significant impact on shorter timescales (Martin et al., 2010; de Sá et al., 2017, 2019). Biomass
burning events, both for land clearing as well as pasture and cropland maintenance, can produce particles at high
number and mass concentrations. Increased urbanization in the Amazon, for example the city of Manaus, Brazil, with
a 2017 population of 2.1 million, represents a large emission source of both gases and particles and has led to increased
regional transportation infrastructure and resulting increases in oxides of nitrogen (NOx) (IBGE, 2017). The latter will
have important implications on the reactive pathways of BVOCs and the formation of secondary organic aerosol
(SOA) (de Sá et al., 2018). With the opportunity to observe aerosol particles under pristine conditions, combined with
the presence of growing urban centers and increased land use change that represent significant regional sources of
oxidants and other key trace gases, this region presents opportunities to understand both past and future drivers of
atmospheric chemistry and climate.
Aerosol properties in the Amazon basin show a seasonal dependence, reflecting seasonal variability in emissions and
deposition. During the wet season (December through March), the region is dominated by natural emissions, as
accumulation mode (particle diameters between 0.1 and 2.5 µm) and coarse mode (diameters above 2.5 µm) particles
tend to be lower in concentration due to wet deposition (Andreae, 2009). In the wet season, ambient particle number
concentrations often represent pristine, near- natural concentrations and are in the range of 300-600 cm$^{-3}$ (Zhou et al.,
2002). Previous measurements of particle number-size distributions in Amazonia during the wet season show ultrafine
particles are present intermittently, most likely linked to times of local pollution events, while both Aitken and
accumulation mode are continuously present (Zhou et al., 2002). While the wet season episodically experiences high
particle number concentrations, the dry season (June through September) experiences higher number concentrations
most of the time, which can alter cloud microphysics, radiative effects, and the hydrological cycle (Andreae et al.,
2002, 2004; Rcia et al., 2000). While it was previously thought that particle composition during the dry period is
dominated by biomass burning, recent measurements of sub-micron particle (PM$_1$) composition show a larger
influence from BVOCs due to decreased wet deposition, resulting in positive feedbacks on oxidants and emissions
(de Sá et al., 2019). Seasonal variations of isoprene, sesquiterpenes and monoterpenes have been measured, with
higher mixing ratios in the dry season (Alves et al., 2016). Additionally, with the lack of rainfall, in-basin pollution
may be more prevalent, especially in areas downwind of cities and settlements (Martin et al., 2010).
Unlike in other forested regions, particles with a diameter smaller than 30 nm are rarely observed over the Amazon
basin, suggesting that new particle formation events seldom occur near the ground (Martin et al., 2010). In other
regions, new particle formation has been seen to occur during the daytime under sunny conditions, suggesting that
both boundary layer dynamics and photochemistry are important factors (Bzdek et al., 2011). Rizzo et al. (2018)
recently analyzed four years of particle size distributions acquired at the TT34 tower site located 60 km northwest of
Manaus. Regional new particle formation and growth events were detected in only 3% of total days observed, whereas
bursts of ultrafine particles that lasted as least an hour occurred during 28% of the days. Those "burst events" were
equally likely to occur during the daytime as the night, and the authors hypothesized that daytime events were caused
by interrupted photochemical new particle formation, whereas nocturnal events might be due to emissions and/or
fragmentation of primary biological particles. Recent airborne observations in the Amazon suggest that particle
nucleation and growth can be initiated in the upper troposphere, with upwelling air masses transporting reactants into
the free troposphere and downwelling air masses transporting aerosol particles and condensable compounds back into
the boundary layer where particles can continue to grow via condensation and coagulation (Andreae et al., 2018; Fan
et al., 2018; Wang et al., 2016). Once formed, ultrafine particles can be key participants in a variety of atmospheric
processes. One example of this is the subject of a recent study by Fan et al. (2018), who have suggested that ultrafine
particles can increase the convective intensity of deep convective clouds. High concentrations of ultrafine particles,
when present with high water vapor concentrations that are typical in the Amazon atmosphere, can form high
concentrations of small cloud droplets that release latent heat and thereby result in more powerful updraft velocities.
While recent research is providing some clarity on the origin, transport, and climate impacts of ultrafine particles in
the Amazon, very little is known about the chemical composition of these particles. Globally, measurements show a
major component of atmospheric ultrafine aerosol are organic compounds produced from BVOC oxidation (Bzdek et

al., 2011; Riipinen et al., 2012; Smith et al., 2008; Smith and Rathbone, 2008). Many of these direct measurements of the composition of atmospheric ultrafine particles have been performed using the Thermal Desorption Chemical Ionization Mass Spectrometer (TDCIMS) (Voisin et al., 2003). For example, TDCIMS measurements performed outside of Mexico City attribute about 90% of the growth of freshly nucleated particles to oxidized organics (Smith et al., 2008). In the Boreal forest of Finland, the contribution of oxidized organics is close to 100% and an analysis of composition suggests that marine emissions can play an important role in that process (Lawler et al., 2018). For the smallest particles measureable by TDCIMS, with diameters from 8 to 10 nm, between 23% to 47% of the constituents may be derived from organic salt formation, a reactive uptake mechanism that requires the presence of strong bases such as gas phase amines (Smith et al., 2010).

Similar to other parts of the world, particles in the Amazon basin are typically composed of 70-80% organics by mass in both the fine and coarse size ranges (Graham et al., 2003). The composition of ultrafine particles has not been directly measured, although one study has proposed the major component could be oxidized organics that have condensed onto potassium salt-rich primary particles emitted from active biota (Pöhlker et al., 2012). An understanding of the origin and chemical composition of ultrafine particles in the Amazon gives insight into their formation and growth processes. To improve upon modelling the coupling of chemistry and climate in this sensitive region, incorporating accurate representations of particle formation and growth processes is required.

The most recent, and currently the largest, field campaign to study the Amazon atmospheric chemistry and cloud processes was the Observations and Modeling of the Green Ocean Amazon (GoAmazon2014/5), which took place outside of Manaus, from 1 January 2014 to 31 December 2015 (Martin et al., 2016). Two intensive observational periods (IOPs) were carried out during GoAmazon2014/5, corresponding to wet and dry seasons in 2014. This manuscript explores the chemical composition of ultrafine particles observed by the TDCIMS during IOP1, which took place from February 1 to March 31, 2014. Specifically, we focus on ten consecutive days that experienced air masses from both remote, primarily forested regions, as well as from the large metropolitan region of Manaus. This study investigates the influence of anthropogenic and biogenic emissions on the chemical composition of ultrafine particles in this region, from which one can infer the chemical processes that led to the formation and growth of ambient ultrafine particles in this region. The time evolution of select compounds in ambient ultrafine particles is analyzed, and compared to AMS measurements, using Principal Component Analysis (PCA), in order to gain additional insights into the contribution of various emission sources to ultrafine particle composition.

**2. Methodology**

**2.1 T3 Site Description**

All data presented were collected at the T3 site (3.2133 $^{o}$S, 60.5987 $^{o}$W), located 70 km west of Manaus, Brazil, during the GoAmazon2014/5 campaign (Martin et al., 2016). The T3 site is located within pasture land located 10 km northeast of Manacapuru, Brazil. The site included the Atmospheric Radiation Measurement (ARM) Mobile Facility #1 (AMF-1), the ARM Mobile Aerosol Observing System (MAOS), and four modified shipping container laboratories containing instruments deployed by universities and other research organizations.

## 2.2 Thermal Desorption Chemical Ionization Mass Spectrometry

Ambient ultrafine particle composition was characterized using TDCIMS. The TDCIMS is an instrument designed specifically for the measurement of the molecular composition of size-resolved ultrafine aerosol particles (Smith et al., 2004; Voisin et al., 2003). In brief, sampled atmospheric particles are charged by a unipolar charger and are collected via electrostatic deposition on a platinum (Pt) filament over varying collection times. During this campaign, collection times were either for 1 hour or 30 minutes, depending on the anticipated sample mass. Typical sample mass collected on the filament ranged from 10 to 100 ng. After collection, the filament was moved into an atmospheric pressure chemical ionization source region and resistively heated to desorb the particulate phase components. These desorbed components were chemically ionized and detected using a quadrupole mass spectrometer (Extrel Corp.). A zero air generator (Parker Hannifin, model HPZA-3500) provided the source of reagent ions $(H_2O)_nH^+$ and $(H_2O)_nO_2^-$ (n=1-3); TDCIMS operation with these ion chemistries is, referred to as either positive and negative ion mode, respectively. Complete mass spectra of desorbed compounds were obtained at the beginning of IOP1 (Fig. S1) to determine ions with the highest ion abundances. These ions were then measured for the duration of the campaign by operating the quadrupole mass spectrometer in "selected ion mode," in which the quadrupole mass spectrometer rapidly switched among approximately 12 ions to optimize sensitivity with high temporal resolution.

Both positive and negative ion mode chemical analyses were performed during the two IOPs, and are publicly available on the campaign data archive (Smith, 2016). During IOP1, several days of measurements were impacted by intermittent power outages and brownouts. IOP2 was characterized by comparatively lower concentrations of ultrafine particles, which is consistent with prior observations (Martin et al., 2010; Rizzo et al., 2018). Because of this, we focus our analysis on ten consecutive days during IOP1 when instruments were operating consistently. This period also happened to coincide with the arrival of two distinct and consecutive air masses, which allows for more accurate side-by-side comparison of aerosol properties during these periods.

Ambient particles were sampled through a 3 m length of Cu tubing with 0.63 cm inside diameter. The inlet extended 0.5 m above the roof of the laboratory, and was curved downawrd and covered with screen to prevent rain and insects from entering. Ambient particles during GoAmazon2014/5 were not size-selected prior to collection on the filament because of low ambient concentrations. The collection process, however, is inherently dependent on particle mobility (McMurry et al., 2009). In order to determine the size-dependent collection efficiency, tests were run at the start of the campaign by generating and collecting ammonium sulfate particles in the diameter range of 8-90 nm. The size-dependent, TDCIMS sampling collection efficiencies were used to determine the volume mean diameter and estimated mass of each sample, as described in Smith et al. (2004).

## 2.3 Meteorological data and complementary datasets

To complement the TDCIMS dataset, High-Resolution Time-of-Flight Aerosol Mass Spectrometry (AMS; Aerodyne, Inc.) was used to characterize non-refractory compounds in $PM_1$ at the T3 site (ARM, 2018a; de Sá et al., 2018). A 7-wavelength aethalometer was located at MAOS and measured black carbon mass concentration (ARM, 2018b). The planetary boundary layer height (ARM, 2018c), determined using the Heffter number method (Heffter, 1980), was measured at MAOS. A Scanning Mobility Particle Sizer(ARM, 2018d) determined the number-size distributions

spanning the mobility diameter range of 10 - 460 nm. Wind direction, wind speed, relative humidity, temperature and
rainfall were measured at AMF-1(ARM, 2018e). Six hour back-trajectory frequency simulations were determined for
the time period of interest using NOAA HYSPLIT transport model, using the GDAS 1° meteorology (Rolph et al.,
2017; Stein et al., 2015).
**2.4 Principal Component and Hierarchical clustering analyses**
Principal Component Analysis (PCA) was performed using the "princomp" function of the R statistical software
package (R, 2011). A hierarchical cluster analysis was performed using Ward's averaging method in the "hclust"
function in R. Ward's minimum variance method of hierarchical clustering was used, which groups species within the
same cluster to minimize the total variance (Wilks, 2011). The purpose of this analysis is to identify species or groups
of species that may have unique sources, trajectories or other physicochemical characteristics. Cluster analysis was
done for the following TDCIMS negative and positive ion mode species: $C_2H_4N^-$ (*m/z* 42), $C_2H_3O_2^-$ (*m/z* 59), $HSO_4^-$
(*m/z* 97), $Cl^-$ (isotopes *m/z* 35 and 37), $HC_2O_4^-$ (*m/z* 89), $NH_4^+(H_2O)$ (*m/z* 36), $K^+$ (*m/z* 39 and 41), $C_3H_{10}N^+$ (*m/z* 60),
$C_5H_7O^+$ (*m/z* 83), $C_5H_8NO^+$ (*m/z* 98), and $C_7H_9O_2^+$ (*m/z* 125) and the following AMS species: organic, ammonium,
nitrate, sulfate and chloride. A separate cluster analysis was performed for quality assurance and demonstrated that
the three clusters presented in Section 3.3 are statistically significant and different from one another.
**3. Results and Discussion**
**3.1 Meteorological Data and Classification of Air Masses**
The ten consecutive days that are the focus of this study can be characterized by two distinct air mass types, as
determined from meteorological data and AMS-derived PMF factors (de Sá et al., 2018). The first period, referred to
as the "anthropogenic period," was from 14 March until mid-morning 19 March and the second period, the
"background period," was from mid-morning 19 March until 24 March. The AMS-derived biomass burning factor
(BBOA), associated with levoglucosan, and anthropogenic-dominated factor (ADOA), associated with mass fragment
91 or "91fac" ($C_7H_7^+$), were as much as three times larger during the anthropogenic period than background period
(de Sá et al., 2018). Anthropogenic influence during this campaign, as determined using ADOA, most strongly
resembled cooking emissions. Correlations between the ADOA factor, cooking emissions, aromatics like benzene,
toluene and xylene and increased particle counts verify the link to anthropogenic influence from Manaus (de Sá et al.,
2018). The particle number-size distribution, shown in Figure 1, for the anthropogenic period saw higher number
concentrations of particles over the diameter range of 10 –100 nm ($N_{100}$). Particle size distributions for the background
period were comparable to previous measurements in the Amazon basin, featuring a bimodal distribution with peaks
at roughly 50 nm and 150 nm and peak concentrations of approximately $10^3$ particles $cm^{-3}$ (Fig S2) (Artaxo et al.,
2013; Gunthe et al., 2009; Pöhlker et al., 2016; Rizzo et al., 2018). The average total mass concentration as determined
by the AMS for the anthropogenic period was $2.5 \pm 0.9$ µg/m³. The T3 site experienced approximately four hours of
rain on 19 March ending at about noon UTC (all times are presented as UTC time, which is four hours ahead of local
time) and the first and only new particle formation event of this ten-day period was observed. After this event on 19
March, number concentrations of particles were, on average, much lower than the prior period. The average total mass

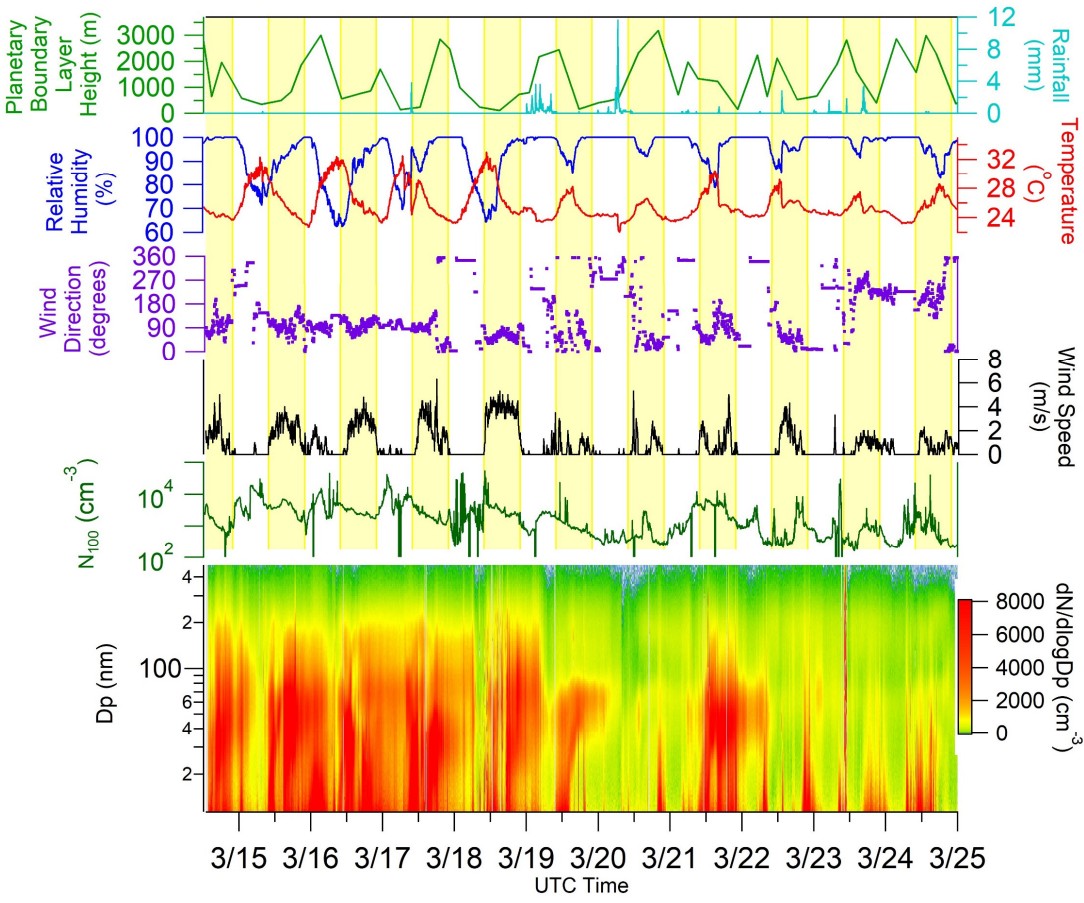

**Figure 1:** Meteorological data from the T3 site, showing planetary boundary layer height (green), rainfall (light blue), relative humidity (dark blue), temperature (red), wind direction (purple), wind speed (black) and total number concentration of sub 100 nm particles ($N_{100}$, dark green). The highlighted yellow bars signify daylight hours (10:00-22:00, UTC time). The particle number- size distribution contour plot shows size distribution function (molecules cm$^{-3}$) for particles sizes between 10 nm and 400 nm.

concentration for the background period was determined to be $1.2 \pm 0.8$ µg/m$^3$. A similar trend in total mass
concentration between background and polluted conditions was observed during the Southern Oxidant and Aerosol
Study (SOAS), where larger particle mass concentrations were observed during times with polluted air mass influence
and, when followed by a period of rainfall, smaller mass concentrations were observed (Liu and Russell, 2017).
Occasional rainfall was seen during the background period, resulting in wet deposition of aerosol particles.
Addtionally, a backtrajectory analysis, presented next, provides a more likely reason for these two distinct periods.
Wind direction data shown in Figure 1, as well as NOAA HYSPLIT data shown in Figure 2, suggest a reason for the
two distinct periods. Back-trajectories show that air masses during the anthropogenic period either pass through
Manaus or south of Manaus prior to arrival at the T3 site. During this period, air masses most frequently passed over
the main roadway that connects Manaus with Manacapuru, a neighboring city with a population of 93,000. Along this
roadside are homes, agriculture and brick kilns, all of which contribute to local gas and particle emissions. In contrast,
during the background period, air masses arrived at the T3 site most frequently from the north east and west. Air

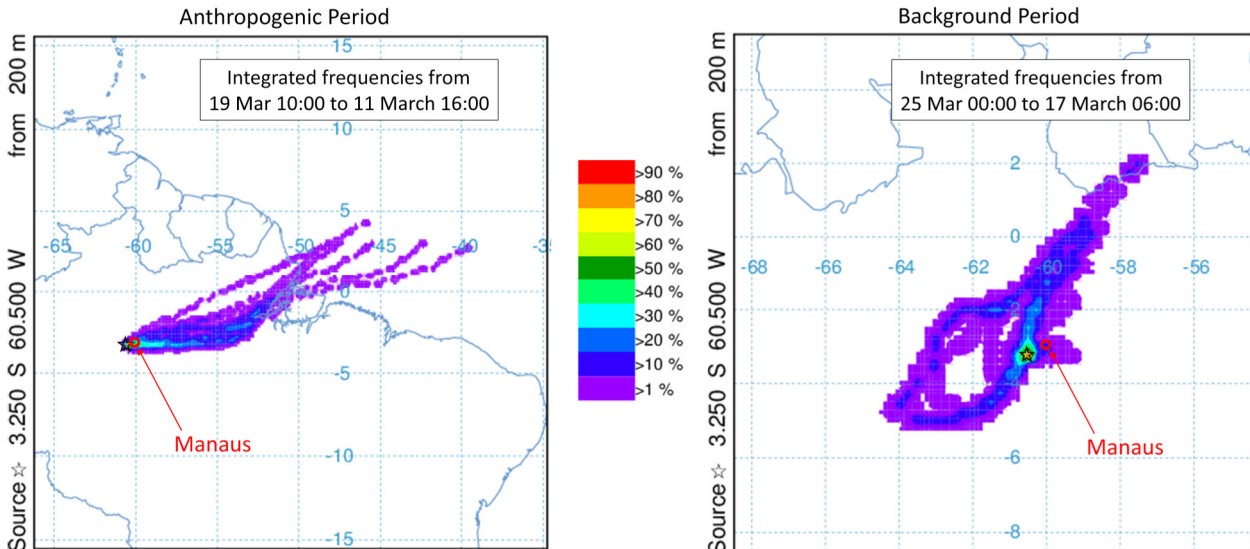

**Figure 2:** Back trajectory frequencies performed using HYSPLIT, showing the different air masses that travel to the T3 site during the anthropogenic period and background period. For each period, twenty trajectories were used to determine integrated frequencies spanning the five days of each period (14 Mar-19 Mar for the anthropogenic and 20 Mar-25 Mar for the background period). Each trajectory duration was for 72 hours. The color scale indicates the frequency of which air masses pass over that area, with the warmer color being more frequently passed over.

masses that were measured at the site typically originated from densely forested regions northeast to west of Manaus. Less frequent were periods where air masses reaching the site originated from east and were influenced by the Manaus metropolitan area. For example, during the evening of 21 March there was a period of increased number concentration and, as winds were quite stagnant at night, it is possible that a local emission source could have impacted the site during that period. Wind direction on this day corresponded with air masses arriving to the T3 site from the Manaus area.

Estimated masses of ultrafine particles sampled by the TDCIMS were determined and compared for the two periods (Fig. S3). During the anthropogenic period there was no distinct diurnal pattern observed, with an average of ~100 ng/sample. This lack of a diurnal pattern in the sampled particles suggests that sources or processes that are responsible for these particles could have persisted throughout the day and night or could be from different processes that persisted both day and night. In contrast to this, the background period has a diurnal peak in estimated mass collected between 18:00 to 22:00 UTC, with sampled masses of ~70 ng/sample. The minimum sample sizes occurred in the early morning where averages reached as low as 16 ng/sample. Peaks in collected mass during the early afternoon could be linked to photochemically produced sources and appear to be unique to the background period. Assuming that the background contribution to the mass of particles remains constant between each time period, the average mass loading of ultrafine particles increased by a factor of 3 due to anthropogenic influence (Fig. S3).

**3.2 Ultrafine particle chemical composition**

The five most abundant negative ions, as observed in full mass spectra (Fig. S1) taken at the start of the wet season campaign, are attributed to $CNO^-$ (cyanate, *m/z* 42), $C_2H_3O_2^-$ (acetate, *m/z* 59), $HSO_4^-$ (bisulfate, *m/z* 97), $Cl^-$ (chloride, isotopes *m/z* 35 and 37) and $HC_2O_4^-$ (hydrogen oxalate, *m/z* 89). The six most abundant positive ions measured were

attributed to $NH_4^+(H_2O)$ (ammonium hydrate, *m/z* 36), $K^+$ (potassium, isotopes *m/z* 39 and 41), $C_3H_{10}N^+$ (trimethyl
ammonium, *m/z* 60), $C_5H_7O^+$ (protonated 3-methylfuran, *m/z* 83), $C_5H_8NO^+$ (*m/z* 98), and $C_7H_9O_2^+$ (*m/z* 125). We will
refer to $C_5H_8NO^+$ (*m/z* 98), and $C_7H_9O_2^+$ (*m/z* 125) collectively as "other" in our positive ion mode analysis as these
were minor components. The major isotopes of chloride were measured in order to understand the role chloride may
have had on particle formation, with potential influence from marine aerosol and fungal spores (Pöhlker et al., 2012).
Potassium (isotopes *m/z* 39 and 41) was measured during positive ion mode analysis to determine the potential
influence of potassium-rich primary biological particles (China et al., 2016; Pöhlker et al., 2012). Additionally,
potassium-rich particles have been linked to biomass burning, as potassium is found to be associated with soot carbon
(Andreae, 1983; Pósfai et al., 2004). Mass-normalized ion abundances, defined as ion abundance divided by collected
sample mass, for the five most abundant negative ions displayed similar diurnal patterns within each period. During
the anthropogenic period, peaks in mass-normalized ion abundance were observed for all measured species between
6:00-8:00 and 16:00-18:00. For the background period, there was no sharp peak observed between 16:00-18:00 for
any of the five measured species, but peak in the diurnal pattern between 6:00-8:00 for *m/z* 42, *m/z* 59 and *m/z* 89 (Fig.
S4). Diurnal trends in mass-normalized ion abundances give little insight, per se, into sources of individual ions, but
it is interesting to note that ion abundances are typically the lowest when sample mass is largest. A potential reason
for this is that TDCIMS is not sensitive to the specific compounds present in these ultrafine particles when the mass
loading is highest. This could be true, for example, if refractory black carbon is the main constituent during the period
of highest sampled mass, as chemical ionization would be unable to detect these compounds. Since the diurnal patterns
of all individual ions are similar, a comparison of ion fractions, defined as ion abundance divided by the sum of the
total ion abundances measured at the time of analysis, provides a measurement of ion concentration in collected
particles and shows distinct differences between the background and anthropogenic periods.
Figure 3a shows the trend in ion fraction for five most abundant negative ions and four most abundant positive ions
during the ten-day period of analysis. During the anthropogenic period, the observed bisulfate ion (*m/z* 97) fraction
was larger than during the background period. Of the ions measured, bisulfate is the predominant indicator of urban
influence. The bisulfate anion has been previously noted in TDCIMS analysis as a stable ion formed from the thermal
desorption of particulate sulfate (Voisin et al., 2003), and it is likely that emissions from Manaus could serve as the
major source for sulfate found at the T3 site. Thus as air masses during the anthropogenic period primarily travelled
from, or south of, Manaus, bisulfate is expected to have a higher measured ion fraction. Additionally, in-basin
emissions of various gaseous precursors like dimethyl sulfide and hydrogen sulfide could contribute to particulate
sulfate of non-anthropogenic origin, as bisulfate was measured during the whole ten-day period of interest, even
without observed direct influence from Manaus. When the bisulfate ion was the largest of the negative ions, the largest
fractions of ammonium (*m/z* 36) and trimethyl ammonium (*m/z* 60) in the positive ion mode were observed as well.
Additionally, the largest chloride (*m/z* 35) signal was observed at the beginning of this period, reaching a maximum
of about 10% of the total ion fraction on 14 March. During the background period, the ion fraction of hydrogen oxalate
(*m/z* 89) remained relatively constant, averaging 31% ± 5% of the total ion fraction. Diurnal patterns of these ion
fractions, shown in Figure 3b, show small diurnal variations for most of the observed ions. The diurnal pattern of *m/z*
42 (cyanate) peaks between 10:00 and noon and both *m/z* 59 and *m/z* 89 show slight decreases between 10:00 and

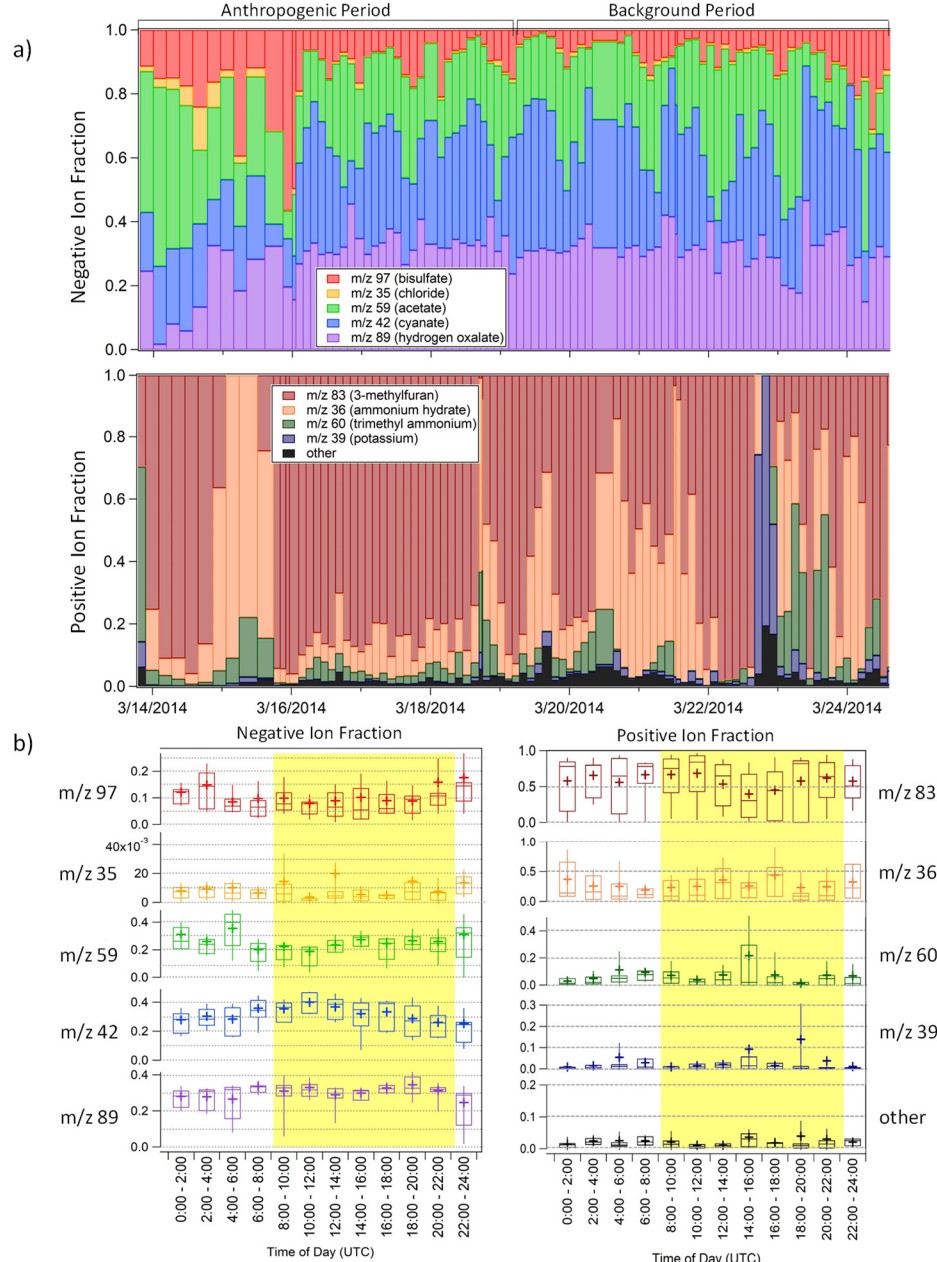

**Figure 3:** a) The negative ion fraction and positive ion fraction shown over the ten-day period of interest. b) Diel patterns of the five measured negative ions shown and of the four major positive ions, "other" refers to sum of fractions of $m/z$ 125 and $m/z$ 98. The crosses are average values, the boxes show 25th and 75th percentiles as well as medians, and the whiskers show maximum and minimum values. Signals are averaged between the two-hour time blocks noted. Highlighted region denotes daylight hours.

noon, as well. Roughly 70% of measurements over both periods had potassium ($m/z$ 39 and 41) ion fractions less than
or equal to 20% of the total positive ion fraction, with few "potassium episodes" of higher abundance observed.
Interestingly, $m/z$ 42 was the most abundant ion present in TDCIMS spectra. Due to its even mass-to-charge ratio, this
ion almost certainly contains nitrogen. This ion distinguishes itself from other detected compounds by a peak in ion
fraction during the morning (Figure 3b). Prior TDCIMS measurements during the 2006 MILAGRO campaign in the
Mexico City Metropolitan Area, detected $m/z$ 42 as a major ion fragment in sub-20 nm diameter particles; that ion

was identified as cyanate (CNO⁻), which may be linked to isocyanic acid from biomass burning or industrial processes (Smith et al., 2008). The *m/z* 42 fragment observed in this study is not likely of anthropogenic origin since this ion was observed during very clean periods when we expect anthropogenic emissions and biomass burning to be low. In addition, TDCIMS-measured *m/z* 42 during the dry season did not show an increase in ion intensity relative to the wet season (Smith, 2016), which one might expect if this ion were sourced to biomass burning. We hypothesize that this ion is cyanate (CNO⁻) which we associate with organic nitrogen related to aerosol formation from biogenic emissions of VOCs. Natural emissions of amino acids, water soluble organic species, and other proteinaceous biogenic material have been measured in the gas phase, particle phase and in precipitation across the globe, and have been estimated to account for as much as half or more of the bulk aerosol composition over the Amazon basin (Artaxo et al., 1988, 1990; Kourtchev et al., 2016; Mace et al., 2003; Zhang and Anastasio, 2003). While all prior field measurements in the Amazon basin have been made on particles larger than those measured in this study, similar sources may influence ultrafine particle composition. If true, these observations suggest that organic nitrogen compounds play a crucial role in both ultrafine particle formation as well as growth to large particles, which make this mechanism for particle growth climatologically important in this region.

Of the measured positive ion species, *m/z* 83, linked to 3-methylfuran or other C5 oxidized volatile organic compound, dominated the ion fraction in ultrafine particles. Methylfuran has been observed to be produced as a thermal decomposition product of isoprene-derived SOA via AMS measurements (Allan et al., 2014), a process that would likely also occur during TDCIMS analysis. Airborne observations in the Amazon suggest that isoprene SOA can be formed in the boundary layer under certain conditions, which is confirmed by these observations (Allan et al., 2014). Since this ion is a marker of isoprene epoxydiol (IEPOX) species present in the particle phase, this confirms a role for isoprene and isoprene derivatives in the growth of ultrafine particles. Little variability in the diel pattern for m/z 83 is observed, similar to other particle phase measurements of IEPOX derivatives reported for the GoAmazon2014/5 campaign by Isaacman-Vanwertz, et al. (2016). In that study, weak diurnal patterns for particle phase isoprene oxidation products were also observed, even while gas phase concentrations of these species increased in the afternoon. It is important to note that this ion dominates the positive ions fraction during both the anthropogenic and background influenced periods. Times that experienced lower fractions of *m/z* 83 had increased fractions of ammonium and trimethyl ammonium, which also coincided at times with larger amounts of measured bisulfate in the negative ions. The presence of larger fractions of particulate ammonia and amines at times with less influence from isoprene-derived species could indicate that both organic salt formation and uptake of isoprene-derived products are possible mechanisms of ultrafine particle growth. The importance of organic salt formation in growth is consistent with prior TDCIMS measurements (Smith et al., 2010), although a quantitative comparison cannot be made since this current study focuses on sub-100 nm diameter particles whereas the prior study focused on size-resolved sub-15 nm ambient particles. One period of elevated potassium ion ratio was observed at the end of the day on 22 March. To differentiate between potential sources of potassium in these ultrafine particles, whether it be of primary biological or biomass burning influence, mass concentrations of black carbon during this ten-day period of interest were used to examine the extent of influence of biomass burning on the presence of potassium (Fig. S5). During the anthropogenic period, with significantly elevated concentrations of black carbon, minimal potassium fraction was measured. At times

of low black carbon mass concentrations during the background period, like on 20 March, there was some fraction of
potassium observed. During the period of highest fraction observed on the night of 22 March, there were slightly
elevated mass concentrations of black carbon. While partially elevated black carbon mass concentrations on 22 March
may be connected to the large potassium ion fraction, at times with even more significant biomass burning influence,
there was minimal potassium. The larger fraction of potassium observed during the background period, as opposed to
the anthropogenic period, may be connected to potassium rich biological particles or the rupturing of biological spores
(China et al., 2016; Pöhlker et al., 2012). Of all wet season TDCIMS measurements during GoAmazon2014/5, roughly
14% of measurements had potassium fractions greater than 0.1 (Fig. S6). Air masses on the evening of 22 March were
traveling steadily from the Manaus area and coincided with about 5 mm of rain. High ambient concentrations of
biological particles that could be sources of potassium are often associated with rainfall events (China et al., 2016).
Rupturing of fungal spores, leading to the production of sub-100 nm fragments, was observed to occur after long
exposures (above 10 hours) of high relative humidity and subsequent drying, similar conditions to those on 22 March.
**3.3 Multivariate analysis of TDCIMS and AMS data**
Principal Component Analysis (PCA) was performed on TDCIMS and AMS measurements to provide insights into
the possible drivers for ultrafine particle formation. Figure 4 shows the results of this analysis. In these plots, positive
correlations are shown in blue, while negative correlations are shown in red. The intensity of the color and eccentricity
of the ellipse is an indication of the degree of correlation. Pale-colored circles (eccentricity approximately 0) show
little to no correlation, narrow ellipses with a positive slope and darker blue color illustrate strong positive correlations
and narrow ellipses with a negative slope and darker red color show strong negative correlation.
Hierarchical clustering of these measurements results in three main clusters of related particle constituents. This
represents a series of clusters where the species within each cluster covary, therefore indicative, in this work, of similar
particle characteristics, processes or sources. The first, labeled "Cluster 1" on Figure 4, grouped TDCIMS-derived
cyanate (*m/z* 42), acetate (*m/z* 59) hydrogen oxalate (*m/z* 89), trimethyl ammonium (*m/z* 60) and 3-methylfuran (*m/z*
83); the second, labeled "Cluster 2," clustered well known co-varying AMS derived constituents (Ulbrich et al., 2009);
the third, labeled "Cluster 3" associated AMS-derived chloride with TDCIMS-derived chloride (*m/z* 35), bisulfate
(*m/z* 97), ammonium hydrate (*m/z* 36) and potassium (*m/z* 39). The hierarchical clustering approach independently
grouped and separated AMS measurements from TDCIMS measurements. While both represent composition
measurements of the aerosol population, the differences between the size ranges of particles measured by AMS and
TDCIMS techniques would lead to the anticipated differences in clustering. Comparing mass distributions estimated
by size distribution measurements, the presence of particles larger than 100 nm would have a more significant
contribution to the measured mass concentrations by AMS. In contrast, the TDCIMS only measures sub-100 nm
particles, representing a minor contribution to the total mass concentration. This observed separation between the
clustering of AMS and TDCIMS measurements reinforces the importance of direct measurements of ultrafine
particles, as opposed to bulk composition, in accessing the species and mechanisms responsible for new particle
formation.

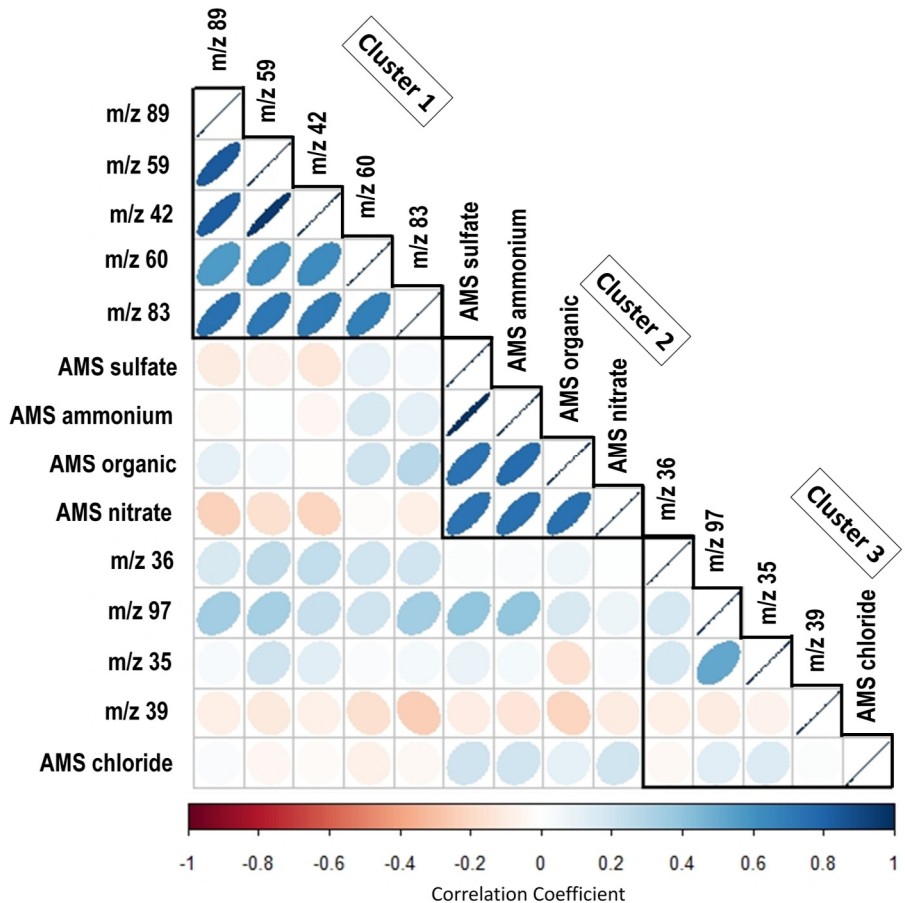

**Figure 4**: Principal Component Analysis (PCA) of TDCIMS and AMS data. Refer to text for details on the interpretation of these plots. PCA results in which species are grouped into hierarchical clusters, with clusters denoted within weighted black lines. Species are ordered by decreasing correlation to the first principal component from the top to bottom. TDCIMS chemical assignments for fragments are m/z 89 (hydrogen oxalate), m/z 59 (acetate), m/z 42 (cyanate), m/z 60 (trimethyl ammonium), m/z 83 (3-methylfuran), m/z 36 (ammonium hydrate), m/z 97 (bisulfate), m/z 35 (chloride) and m/z 39 (potassium).

With respect to PCA performed on the two datasets, Cluster 1, which includes TDCIMS fragments typically linked to
organic species (*m/z* 59, 89, 83) and nitrogen species discussed previously (*m/z* 42 and 60), explains most of the
variance and has the highest correlation with the first principal component. These species' high correlation with each
other indicate similar sources, most of which can be associated with BVOC emissions. A prior TDCIMS laboratory
study linked the acetate ion fragment (*m/z* 59) to particulate carboxylic and dicarboxylic acids (Smith and Rathbone,
2008), which have been linked to the photochemical oxidation of both biogenic and anthropogenic compounds
(Winkler et al., 2012). During the wet season in the Amazon basin, specific dicarboxylic acids and tricarboxylic acids
have been identified and proposed to have been formed from the oxidation of semi-volatile fatty acids and terpenes
(Kubátová et al., 2000). Hydrogen oxalate, measured as *m/z* 89, was one of the two most abundant organic ions
measured in ultrafine particles at both an urban and rural site in Helsinki, Finland (Pakkanen et al., 2000). Hydrogen
oxalate was noted to have relatively constant concentrations in ultrafine particles, similar to observations seen during
the ten-day period of analysis for this study (Figure 3). While Helsinki and the Amazon experience different conditions
and meteorology, oxalate has been observed in both environments, possibly due to the heavy BVOC influence in both

locales. In the positive ion mode, 3-methylfuran, measured as *m/z* 83, has significant correlation to background linked negative ions. These species seem to be generally linked to the oxidation of various BVOCs, whether isoprene, for 3-methylfuran, or other terpenes (Allan et al., 2014). Finally, it should be noted that the clustering of the cyanate (*m/z* 42) with these organic ions provides further evidence that the source of this ion is likely clean, background chemistry rather than from biomass burning. Additionally, TDCIMS-measured cyanate (*m/z* 42) are weakly and negatively correlated to AMS-measured nitrate. During the anthropogenic period (14 March through mid-morning 19 March), higher levels of inorganic nitrate were observed by AMS compared to the organic form (de Sá et al., 2018). This higher mass concentration of nitrate attributed to inorganic nitrate, as opposed to organic nitrate which would be more similar to TDCIMS-measured cyanate, should explain the slight negative correlation between the two.

Hierarchical clustering separates TDCIMS-measured ions into two clusters, with Cluster 3 including TDCIMS-derived bisulfate (*m/z* 97), chloride (*m/z* 35 and 37), ammonium hydrate (*m/z* 36) and potassium (*m/z* 39 and 41). The separation of this cluster suggests that these constituents are linked to different sources or atmospheric processes compared to those in Cluster 1, potentially with an anthropogenic origin as both chloride, potassium and bisulfate have been linked previously to biomass burning and anthropogenic emissions, respectively (Allen and Miguel, 1995; Martin et al., 2010; Voisin et al., 2003). As noted previously, the bisulfate anion is stable ion formed from the thermal desorption of particulate sulfate (Voisin et al., 2003) and it is likely present in ultrafine particles via pollution emissions from Manaus. However, in-basin emissions of sulfate gaseous precursors, like dimethyl sulfide and hydrogen sulfide, could be linked to the measured bisulfate fraction during the entire ten-day period with anthropogenic sources of sulfate increasing this background level during the anthropogenic period. In-basin chloride emissions could come from both biomass burning of common regional vegetation and long range transport of marine ultrafine particles from the Atlantic Ocean under influence of the Trade Winds (Allen and Miguel, 1995; Martin et al., 2010). The clustering of AMS chloride with TDCIMS species in Cluster 3 might suggest similar sources of chloride in both ultrafine particles and PM2.5. However, it is worth noting that AMS chloride also very weakly correlated with the other species measured by the AMS. For this reason, its inclusion in this cluster indicates both that AMS chloride is similar to TDCIMS-derived Cluster 3 species and different enough so as not to cluster with the other AMS species. The production of potassium, which is potentially linked to rupturing of fungal spores and biomass burning, would have little correlation to other measured TDCIMS species, as the mechanism for the production of potassium is independent of SOA formation mechanisms.. This ion is not generally associated to constant background sources, like TDCIMS species observed in Cluster 1, and may be associated with potential anthropogenic sources, like bisulfate and chloride seen in Cluster 3. The clustering of TDCIMS ion abundances into two clusters suggests different sources and processes for these species, as there is little correlation between the species present in Cluster 1 to those present in Cluster 3.

**4. Conclusion**

The chemical composition of ultrafine particles in the Amazon basin, as measured during the GoAmazon2014/5, has two distinct influences: sources and processes linked to anthropogenic origin and those related to more natural sources and processes. During periods of heavier anthropogenic influence, higher number concentrations of sub-100 nm

particles were observed (Figure 1). HYSPLIT back trajectories during the anthropogenic period (Figure 2) not only intersect with the Manaus metropolitan area, but with the main roadway that connects Manaus with the city of Manacapuru. Influence from anthropogenic sources, which during the study period are primarily linked to Manaus metropolitan area emissions, may continuously affect the composition of ultrafine particles observed at the T3 measurement site. Particulate sulfate, measured as the bisulfate ion, was an important and dominant contributor to TDCIMS ion fraction during the anthropogenic period (Figure 3), but was still measured, to a lesser extent, in the background period, suggesting an omnipresent influence. The most abundant negative ion species measured during this campaign, likely related to organic nitrogen species at $m/z$ 42, displayed a consistent morning diurnal peak and was an equally abundant constituent during both the anthropogenic and background periods. The dominance of this ion during both this study and the 2006 MILAGO campaign in the Mexico City Metropolitan area emphasizes the potential role of organic nitrogen in ultrafine aerosol particle formation and underscores the need for further research into the chemical processes and precursors that are responsible for this ion. 3-Methylfuran, measured as $m/z$ 83, was the most dominant fraction observed in the positive ion mode and is likely associated with IEPOX derivatives present in ultrafine particles. The presence of these species emphasizes the important of isoprene oxidation to particle formation in this region. The two different clusters of TDCIMS-derived ions that arise through PCA analysis, of which Cluster 1 explains most of the variance, give additional insight into the sources and processes that influence the ultrafine particle population in this part of the Amazon basin. As hierarchical clustering separates TDCIMS-derived organic species from TDCIMS-derived sulfate and chloride, this suggests these species are present in the particle from different sources and/or processes. A third cluster separates AMS-measured compounds from those detected by TDCIMS, which emphasizes the unique characteristics of ultrafine particles compared to bulk aerosol particles. The lack of correlation between the two TDCIMS-derived clusters supports the observation that anthropogenic emissions and processes each have a unique role to play in ultrafine particle formation and growth in the Amazon basin.

**Author contributions**

JNS, PA, STM, OVB, RdS, and JT designed the measurement campaign and JNS, MJL, JO, SSdS carried out measurements. HSG performed data analysis, assisted by JNS and AC. HSG prepared the manuscript with contributions from all co-authors.

**Competing interests**

The authors declare that they have no conflict of interest.

**Acknowledgements**

Institutional support was provided by the Central Office of the Large Scale Biosphere Atmosphere Experiment in Amazonia (LBA), the National Institute of Amazonian Research (INPA), and Amazonas State University (UEA) and the local Research Support Foundation (FAPEAM/GOAMAZON). We acknowledge support from the Atmospheric

Radiation Measurement (ARM) Climate Research Facility, a user facility of the United States Department of Energy, Office of Science, sponsored by the Office of Biological and Environmental Research, and support from the Atmospheric System Research (ASR, DE-SC0011122 and DE-SC0011115) program of that office. JS acknowledges support from a Brazilian Science Mobility Program (Programa Ciência sem Fronteiras) Special Visiting Researcher Scholarship. PA acknowledges funding from FAPESP – Fundação de Apoio à Pesquisa do Estado de São Paulo, Grants number 2017/17047-0, 2013/05014-0 and 2014/50848-9.

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
