# Peer review of "Chemical composition of ultrafine aerosol particles in central Amazonia during the wet season"

_Atmospheric Chemistry and Physics, 2019_

## Referee Comment (RC1) · Anonymous Referee #1 · 20 Jun 2019

Glicker et al. presents the chemical composition of ultrafine particles (<100 nm) during two distinct periods of the GoAmazon campaigns. The first period was characterized by air masses passing through a large, urban area and the second by air from the forest (i.e. background). The authors used a thermal desorption chemical ionization spectrometer (TDCIMS) to measure the chemical composition of particles found in and/or produced from these two distinct air masses. Their results indicate that ultrafine particles during the anthropogenic period contained more bisulfate and ammonium+ trimethyl ammonium. During the background times, the particles contained isoprene-derived organic compounds. Organic nitrogen compounds were found to be important in both time periods, indicating their importance in particle formation and growth of ultrafine particles. Comparison of the TDCIMS results with the AMS indicates key

differences in ultrafine composition (and consequently, chemistry) compared to larger particles (>100 nm). Overall, this study helps address the clear gap in knowledge of ultrafine particle composition. In addition, their results show that isoprene chemistry plays a role in new particle formation in the Amazon Rainforest and likely preindustrial times.

This study is easy to follow and the topic is appropriate for ACP. I have a few minor comments the authors should address but recommend this manuscript be published.

Page 2, line 49: "impacts of particle physical and. . ." awkward phrasing

Page 2, line 50: "Uncertainty would be aided. . ." awkward phrasing

Page 2, line 65: What do the authors mean by "large area sources"?

Page 3 line 80: larger should be higher

Page 3, line 93: "in only 3% of the days"

Page 5. Line 172: It is not clear how the authors used collection efficiency of the sampling line to determine mass of each sample. Is size dependent charge fraction from the unipolar charger taken into account? Please expand on this description a bit more to make it clearer.

Page 7 line 218: "air masses often also passed over"

Page 7 line 221: might be better to clearly state that air masses from the northeast to northwest are from the forest.

Page 8 line 226: ∼100 ng/sample (and figure S2) From this sentence and the SI figure, it seems like the collection time was 2 hours and was the same for each sample. If the sampling times for the TDCIMS varied (as the TDCIMS description suggested on Page 5 line 149 and several of the other figures), how is the reader suppose to compare mass loadings per sample between the background and anthropogenic periods? Ultimately, the units of ng/sample are difficult to compare to other studies if sampling times and

volumetric flow rates are not known. It would be worth converting ng/sample to mass concentration.

Page 8 line 247: were measured in order to understand

Page 8 line 247 and line 268: The TDCIMS measured potassium since it is a tracer for primary biological fragments. However, potassium is also a well-known (albeit imperfect) tracer for biomass burning. Figure 3 shows elevated potassium during the anthropogenic period and the tail end of the background period where sulfate fraction increased. Could this be from biomass burning?

Page 10 line 279: Super interesting that 42 m/z was the most abundant ion. The authors attribute this to C2H4N-. Do the authors know what compounds would lead to this ion fragment? Is it possible that 42 m/z showing up in both the anthropogenic and background period could be explained by two compounds or types of compounds? Maybe cyanate contributed to the 42 m/z signal during the anthropogenic period and other organic nitrogen during the background period?

Page 10 line 294: It is a bit strange the authors used PM2.5 observations of organic nitrogen to justify 42 m/z being organic nitrogen in sub 100 nm particles as they later state that ultrafine particles have unique composition compared to larger particles.

Figure 1: It is nearly impossible to read the numbers on the color scale for the particle size distribution color map. Also units of dN/dlogDp should be (cm-3) and not molecules/cmˆ3. Is there a reason the rainfall scale is wide when the amount of rain does not exceed ∼4 mm? Also, the left-hand axes for wind direction and windspeed look like continuations of the observations (i.e. large spike in wind speed and 360 degrees for wind direction). Please make this a bit clearer (and larger so everything is easier to read).

Figure 2: Would be helpful to label Manus on this map. Could be a patch of rocks to the unsuspecting reader. Also please add that pink and red traces are from anthropogenic

[Figure]

period in the caption.

Figure 3: The text in the legend is too small to read. Would be helpful if something was drawn on (a) to indicate when the anthropogenic and background periods were. For (b), consider putting in the m/z for each panel because the reds, greens, blues, and yellows (orange?) look the same.

Figure 4: label for the color scale would be useful, especially since there are no -1 correlations (red) but only light oranges that are harder see.

Figure S1: bisulfate is a clearer marker for anthropogenic influences compared to what?

Figure S3: A legend is needed for each panel/color. Is this negative ion? Or positive?

[Figure]

---

## Referee Comment (RC2) · Anonymous Referee #2 · 30 Jun 2019

I attached my comments as a pdf file.

Please also note the supplement to this comment:
https://www.atmos-chem-phys-discuss.net/acp-2019-299/acp-2019-299-RC2-supplement.pdf
* * *

---

## Referee Comment (RC3) · Anonymous Referee #3 · 2 Jul 2019

*Review of "Chemical composition of ultrafine aerosol particles in central Amazonia during the wet season" by Glicker et al.*

**General Comments:**

This manuscript reports the composition of ultrafine particles during the wet season in central Amazonia as measured by a Thermal Desorption Chemical Ionization Mass Spectrometer (TD-CIMS). The top five abundant ions by signal from each of negative and positive ion modes are reported for a ten-day period representing anthropogenically-influenced and background conditions. The authors find that particulate bisulfate is elevated during the anthropogenic period, though omnipresent, and that organic nitrogen is characteristic of background airmasses. 3-methylfuran (ascribed to IEPOX chemistry) is the dominant component in positive ion mode and interpreted to contribute to new particle growth and formation processes. Finally, the authors find using principal component analysis that ultrafine particle composition can be divided into two clusters, one mostly comprised of organics, and the other comprised of inorganic ions, both distinct from a third cluster with most AMS PM1 measured constituents, indicating unique sources/chemistry for ultrafine and PM1 particles. Overall, this work provides novel measurement of ultrafine particle composition in central Amazonia and would be appropriate for publication in ACP after the following comments are addressed. It is generally written clearly, but lacks some depth in providing additional insight from the measurements. For example, the discussion on PCA analysis could provide more insight into the observed correlations between species/clusters, and as written tends to just reiterate earlier descriptions of the ascribed sources for TDCIMS ion assignments.

**Specific Comments:**

1) Line 60: In addition to Alves et al., 2016, consider adding citation to the following:
   a. Jardine, K. J., Yañez Serrano, a., Arneth, a., Abrell, L., Jardine, A. B., Van Haren, J., Artaxo, P., Rizzo, L. V., Ishida, F. Y., Karl, T., Kesselmeier, J., Saleska, S. and Huxman, T.: Within-canopy sesquiterpene ozonolysis in Amazonia, J. Geophys. Res. Atmos., 116(19), 1–10, doi:10.1029/2011JD016243, 2011.
   b. Jardine, A. B., Jardine, K. J., Fuentes, J. D., Martin, S. T., Martins, G., Durgante, F., Carneiro, V., Higuchi, N., Manzi, A. O. and Chambers, J. Q.: Highly reactive light-dependent monoterpenes in the Amazon, Geophys. Res. Lett., 42(5), 1576–1583, doi:10.1002/2014GL062573, 2015.
   c. Shrivastava, M. K., Andreae, M. O., Artaxo, P., Barbosa, H. M. J., Berg, L. K., Brito, J., Ching, J., Easter, R. C., Fan, J., Fast, J. D., Feng, Z., Fuentes, J. D., Glasius, M., Goldstein, A. H., Alves, E. G., Gomes, H., Gu, D., Guenther, A., Jathar, S. H., Kim, S., Liu, Y., Lou, S., Martin, S. T., McNeill, V. F., Medeiros, A., de Sá, S. S., Shilling, J. E., Springston, S. R., Souza, R. A. F., Thornton, J. A., Isaacman-VanWertz, G., Yee, L. D., Ynoue, R., Zaveri, R. A., Zelenyuk, A. and Zhao, C.: Urban pollution greatly enhances formation of natural aerosols over the Amazon rainforest, Nat. Commun., 10(1), 1046, doi:10.1038/s41467-019-08909-4, 2019.
   d. Yáñez-Serrano, A. M., Nölscher, A. C., Williams, J., Wolff, S., Alves, E. G., Martins, G. A., Bourtsoukidis, E., Brito, J. F., Jardine, K. J., Artaxo, P. and Kesselmeier, J.: Diel and

seasonal changes of biogenic volatile organic compounds within and above an Amazonian rainforest, Atmos. Chem. Phys, 15, 3359–3378, doi:10.5194/acp-15-3359-2015, 2015.

  e. Yee, L. D., Isaacman-Vanwertz, G., Wernis, R. A., Meng, M., Rivera, V., Kreisberg, N. M., Hering, S. V, Bering, M. S., Glasius, M., Upshur, M. A., Bé, A. G., Thomson, R. J., Geiger, F. M., Offenberg, J. H., Lewandowski, M., Kourtchev, I., Kalberer, M., de Sá, S. S., Martin, S. T., Alexander, M. L., Palm, B. B., Hu, W., Campuzano-Jost, P., Day, D. A., Jimenez, J. L., Liu, Y. J., Mckinney, K. A., Artaxo, P., Viegas, J., Manzi, A., Oliveira, M. B., De Souza, R., Machado, L. A. T., Longo, K. and Goldstein, A. H.: Observations of sesquiterpenes and their oxidation products in central Amazonia during the wet and dry seasons, Atmos. Chem. Phys, 18, 10433–10457, doi:10.5194/acp-18-10433-2018, 2018.

2) Lines 73-78: It might be worthwhile to define "ultrafine", "Aitken", "accumulation", and "coarse mode" particles for readers less familiar with these distinctions in Dp ranges.

3) Line 101: Please rephrase "…can have an oversized impact…" as it is not very scientifically clear wording.

4) Line 156: Can the authors also include the MS for positive ion mode? Why was m/z 75 not selected for regular measurement considering its ion intensity is relatively large?

5) Line 160: Please provide additional information in Smith, 2016 under references to make it easier to find.

6) Line 161: Can you specify the threshold for "low concentrations" of ultrafine particles?

7) Figure 1: For size distribution plot, why are the units of intensity for dN/dlogdP in molec/cm^3 rather than #/cm^3 considering that particle concentrations have been discussed earlier in manuscript as #/cm^3?

8) Lines 224-238: This is very interesting analysis. Would the authors be able to infer from this an average % increase in loading on top of "background" conditions that is attributable to anthropogenic influence, assuming that the "background" composition from the March 15-March 19 period is approximately same for the March 20-25 period?

9) Figure S3: Please include figure legends for the ions shown in these diurnal profiles and specify that this is negative ion mode in caption.

10) Lines 264-269: Can the authors clarify if bisulfate ion as indicator of particulate sulfate can also include natural/background sources of sulfate? Since it has been previously established that there are a lot of natural sources of sulfate (e.g. DMS) (Andreae et al., 1990; Andreae and Andreae, 1988) as well as background levels (long-range transport including anthropogenic) (de Sá et al., 2017), would the authors anticipate these sources to be contributing to the majority of the bisulfate anion signal during 19 Mar to 26 Mar 2014?

11) Line 293: Please specify basis of 55-95% of PM as mass basis, etc.

12) Lines 305-306: Based on the diurnal profile of m/z 83 assigned as 3-methylfuran, the authors could better support the claim for IEPOX as a proposed source by comparing with diurnal profiles of gas-phase isoprene oxidation products by PTR-MS (Liu et al., 2016, 2018), particle-phase isoprene oxidation products by SV-TAG Figure 1c, d ((Isaacman-VanWertz et al., 2016), and AMS IEPOX-SOA PMF factor Figure 4b (de Sá et al., 2018) ? Does it make sense for Isoprene + OH → IEPOX to occur 8:00-10:00 UTC and peak, followed by minimum 14:00-16:00 UTC, and then build again?

13) Figure 4: Can the chemical assignments be added after the TDCIMS measured m/z's for ease of chemical interpretation just looking at figure, (e.g. m/z 89 hydrogen oxalate, m/z 59 acetate, etc.)
14) Lines 371-373: Move this explanation of natural bisulfate sources up in manuscript based on Specific Comment regarding Lines 264-269) above.
15) Section 3.3. Authors should include more analysis and interpretation of Figure 3. Can any of these questions below be answered with the PCA analysis:
   a. What do the authors make of the fact that AMS chloride and TDCIMS m/z 35 chloride are in the same cluster despite different size distribution ranges of the two measurement techniques?
   b. Why does TDCIMS measured m/z 42 (organic nitrogen) negatively correlate with AMS nitrate? What implications does this have in terms of sources of organic nitrogen between ultrafine and PM1?
   c. Lines 375-379: Why was AMS-measured K+ not included in the PCA analysis to see if it is distinct/similar to TDCIMS across size distributions?
16) Lines 379-381: Repetitive with lines 365-367.

**Technical Corrections:**

1) Line 56: Delete "is."
2) Line 124: Change "process" to "processes."
3) Lines 131: Change "mass" to "masses."
4) Lines 174-181: Reorder references to ARM, 2018a-d so they appear in alpha order.
5) Line 353: Delete "and" at start of line.
6) Line 395: No need to capitalize "Area."
7) Line 396: Change "underscore" to "underscores"

**References:**

Andreae, M. O. and Andreae, T. W.: The Cycle of Biogenic Sulfur Compounds Over the Amazon Basin 1. Dry Season, J. Geophys. Res., 93(D2), 1487–1497, doi:10.1029/JD093iD02p01487, 1988.

Andreae, M. O., Berresheim, H., Bingemer, H., Jacob, D. J., Lewis, B. L., Li, S.-M. and Talbot, R. W.: The atmospheric sulfur cycle over the Amazon Basin: 2. Wet season, J. Geophys. Res., 95(D10), 16813, doi:10.1029/JD095iD10p16813, 1990.

Isaacman-VanWertz, G., Yee, L. D., Kreisberg, N. M., Wernis, R., Moss, J. A., Hering, S. V., de Sá, S. S., Martin, S. T., Alexander, M. L., Palm, B. B., Hu, W. W., Campuzano-Jost, P., Day, D. A., Jimenez, J. L., Riva, M., Surratt, J. D., Viegas, J., Manzi, A., Edgerton, E. S., Baumann, K., Souza, R., Artaxo, P. and Goldstein, A. H.: Ambient Gas-Particle Partitioning of Tracers for Biogenic Oxidation, Environ. Sci. Technol., 50(18), 9952–9962, doi:10.1021/acs.est.6b01674, 2016.

Liu, Y. J., Brito, J., Dorris, M. R., Rivera-Rios, J. C., Seco, R., Bates, K. H., Artaxo, P., Duvoisin, S., Keutsch, F. N., Kim, S., Goldstein, A. H., Guenther, A. B., Manzi, A. O., Souza, R. A. F., Springston, S. R., Watson, T. B., McKinney, K. A. and Martin, S. T.: Isoprene photochemistry over the Amazon rainforest, Proc. Natl. Acad. Sci., 113(22), 6125–6130, doi:10.1073/pnas.1524136113, 2016.

Liu, Y. J., Seco, R., Kim, S., Guenther, A. B., Goldstein, A. H., Keutsch, F. N., Springston, S. R., Watson, T.

B., Artaxo, P., Souza, R. A. F., McKinney, K. A. and Martin, S. T.: Isoprene photo-oxidation products quantify the effect of pollution on hydroxyl radicals over Amazonia, Sci. Adv., 4(4), eaar2547, doi:10.1126/sciadv.aar2547, 2018.

de Sá, S. S., Palm, B. B., Campuzano-Jost, P., Day, D. A., Newburn, M. K., Hu, W., Isaacman-VanWertz, G., Yee, L. D., Thalman, R., Brito, J., Carbone, S., Artaxo, P., Goldstein, A. H., Manzi, A. O., Souza, R. A. F., Mei, F., Shilling, J. E., Springston, S. R., Wang, J., Surratt, J. D., Alexander, M. L. L., Jimenez, J. L. and Martin, S. T.: Influence of urban pollution on the production of organic particulate matter from isoprene epoxydiols in central Amazonia, Atmos. Chem. Phys, 17(11), 6611–6629, doi:10.5194/acp-17-6611-2017, 2017.

de Sá, S. S., Palm, B. B., Campuzano-Jost, P., Day, D. A., Hu, W., Isaacman-VanWertz, G., Yee, L. D., Brito, J., Carbone, S., Ribeiro, I. O., Cirino, G. G. G., Liu, Y. J., Thalman, R., Sedlacek, A., Funk, A., Schumacher, C., Shilling, J. E., Schneider, J., Artaxo, P., Goldstein, A. H., Souza, R. A. F., Wang, J., McKinney, K. A., Barbosa, H., Lizabeth Alexander, M., Jimenez, J. L. and Martin, S. T.: Urban influence on the concentration and composition of submicron particulate matter in central Amazonia, Atmos. Chem. Phys, 18(16), 1–56, doi:10.5194/acp-18-12185-2018, 2018.

---

## Author Response (AR1)

We thank the reviewers for their comments and feel that incorporating these comments has strengthened our paper. We have addressed the following comments in the sections below and made the appropriate revisions to the manuscript. Reviewer comments are in black text followed by our response in red text. The page and line numbers associated with in-text changes are included in our responses below. The revised manuscript, including markups, and supplemental information have been attached at the end of this document.

Anonymous Referee #1

Glicker et al. presents the chemical composition of ultrafine particles (<100 nm) during two distinct periods of the GoAmazon campaigns. The first period was characterized by air masses passing through a large, urban area and the second by air from the forest (i.e. background). The authors used a thermal desorption chemical ionization spectrometer (TDCIMS) to measure the chemical composition of particles found in and/or produced from these two distinct air masses. Their results indicate that ultrafine particles during the anthropogenic period contained more bisulfate and ammonium+trimethyl ammonium. During the background times, the particles contained isoprene-derived organic compounds. Organic nitrogen compounds were found to be important in both time periods, indicating the importance in particle formation and growth of ultrafine particles. Comparison of the TDCIMS results with the AMS indicates key differences in ultrafine composition (and consequently, chemistry) compared to larger particles (>100 nm). Overall, this study helps address the clear gap in knowledge of ultrafine particle composition. In addition, their results show that isoprene chemistry plays a role in new particle formation in the Amazon Rainforest and likely preindustrial times.

This study is easy to follow and the topic is appropriate for ACP. I have a few minor comments the authors should address but recommend this manuscript be published.

Page 2, line 49: "impacts of particle physical and. . ." awkward phrasing

Sentence has been changed accordingly:
Page 2, line 48-49: "Models are unable to predict the relationships between particle  physico-chemical properties and cloud formation and precipitation (IPCC, 2013)."

Page 2, line 50: "Uncertainty would be aided. . ." awkward phrasing

Sentence has been changed accordingly:
Page 2, line 49-51: "Reducing this uncertainty requires an understanding of the mechanisms by which particles form and grow in the atmosphere, which mostly determine the potential of these particles to serve as cloud condensation nuclei (CCN)."

Page 2, line 65: What do the authors mean by "large area sources"?

Sentence has been changed accordingly:
Page 2, line 64-66: "Increased urbanization in the Amazon, for example the city of Manaus, Brazil, with a 2017 population of 2.1 million, represents a large emission source of both gases and particles and has led to increased regional transportation infrastructure and resulting increases in oxides of nitrogen (NOx) (IBGE, 2017)."

Page 3 line 80: larger should be higher

Sentence has been changed from "larger number concentrations" to "higher number concentrations."

Page 3, line 93: "in only 3% of the days"

Sentence has been changed accordingly:

Page 3, lines 93-97: "Regional new particle formation and growth events were detected in only 3% of total days observed, whereas bursts of ultrafine particles that lasted as least an hour occurred during 28% of the days. Those "burst events" were equally likely to occur during the daytime as the night, and the authors hypothesized that daytime events were caused by interrupted photochemical new particle formation, whereas nocturnal events might be due to emissions of primary biological particles."

Page 5. Line 172: It is not clear how the authors used collection efficiency of the sampling line to determine mass of each sample. Is size dependent charge fraction from the unipolar charger taken into account? Please expand on this description a bit more to make it clearer.

Our procedure for estimating collected sample mass is described in detail in our prior publications. As the reviewer states, we failed to make reference to the prior works and therefore have changed the text as follows:

Page 5, lines 171-173: "The size-dependent, TDCIMS sampling collection efficiencies were used to determine the volume mean diameter and estimated mass of each sample, as described in Smith et al. (2004)."

Page 7 line 218: "air masses often also passed over"

An update to the NOAA HYSPLIT back-trajectories (Fig 2) have been made to include back-trajectory frequencies, so this recommendation has been changed to the following:

Page 8, line 226-229: "Air masses that were measured at the site typically originated from densely forested regions northeast to west of Manaus. Less frequent were periods where air masses reaching the site originated from east and were influenced by the Manaus metropolitan area."

Page 7 line 221: might be better to clearly state that air masses from the northeast to northwest are from the forest.

Sentence has been changed accordingly:
Page 8, line 226-229: "Air masses that were measured at the site typically originated from densely forested regions northeast to west of Manaus. Less frequent were periods where air masses reaching the site originated from east and were influenced by the Manaus metropolitan area."

Page 8 line 226: ~100 ng/sample (and figure S2) From this sentence and the SI figure, it seems like the collection time was 2 hours and was the same for each sample. If the sampling times for the TDCIMS varied (as the TDCIMS description suggested on Page 5 line 149 and several of the other figures), how is the reader suppose to compare mass loadings per sample between the background and anthropogenic periods? Ultimately, the units of ng/sample are difficult to compare to other studies if sampling times and volumetric flow rates are not known. It would be worth converting ng/sample to mass concentration.

The diurnal plot for estimated mass collected shown in figure S2 shows the average mass collected between the two-hour time blocks noted. These are not indicative of two-hour sampling times, as we collected particles for either 1 hour or 30 minutes noted in section 2.2. Estimating mass concentration in air requires many approximations that would result in greater uncertainty in the derived values compared with our current approach. While work is currently underway to reduce these approximation errors, at this time we do not believe that these estimations would be of sufficient accuracy to allow intercomparisons with other studies.

Page 8 line 247: were measured in order to understand

This correction has been made.

Page 8 line 247 and line 268: The TDCIMS measured potassium since it is a tracer for primary biological fragments. However, potassium is also a well-known (albeit imperfect) tracer for biomass burning. Figure 3 shows elevated potassium during the anthropogenic period and the tail end of the background period where sulfate fraction increased. Could this be from biomass burning?

An additional supplemental figure has been added (Fig. S4) showing the positive ion fractions plotted alongside measured black carbon mass concentration. During the anthropogenic period, elevated mass concentrations of black carbon were observed, however, the potassium fraction was near zero. During the background period, there were brief episodes of elevated black carbon mass concentration. At these brief elevated times, we do see some increase in the potassium ion fraction. However, the one big potassium event observed on 22 March is not when the black carbon mass concentration is very high. While black carbon is albiet an imperfect tracer of biomass burning, as noted, it does not seem like the times of elevated potassium ion fraction are linked to high black carbon mass concentration.

The following additions have been added to the paper in the corresponding locations:
Page 9 line 253-255: "Additionally, potassium-rich particles have been linked to biomass burning, as potassium is found to be associated with soot carbon (Andreae, 1983; Pósfai et al., 2004)."

Page 12 starting at line 322: "One period of elevated potassium ion ratio was observed at the end of the day on 22 March. To differentiate between potential sources of potassium in these ultrafine particles, whether it be of primary biological or biomass burning influence, mass concentrations of black carbon during this ten-day period of interest were used to examine the extend of influence of biomass burning on the presence of potassium (Fig. S4). During the anthropogenic period, with significantly elevated concentrations of black carbon, minimal potassium fraction was measured. At times of low black carbon mass concentrations during the background period, like on 20 March, there was some fraction of potassium observed. During the period of highest fraction observed on the night of 22 March, there were slightly elevated mass concentrations of black carbon. While partially elevated black carbon mass concentrations on 22 March may be connected to the large potassium ion fraction, at times with even more significant biomass burning influence, there was minimal potassium. The larger fraction of potassium observed during the background period, as opposed to the anthropogenic period, may be connected to potassium rich biological particles or the rupturing of biological spores (China et al., 2016; Pöhlker et al., 2012)."

Page 15, line 399-402: "The production of potassium, which is potentially linked to rupturing of fungal spores and biomass burning, would have little correlation to other measured TDCIMS species, as the mechanism for the production of potassium is independent of SOA formation mechanisms."

Page 10 line 279: Super interesting that 42 m/z was the most abundant ion. The authors attribute this to C2H4N-. Do the authors know what compounds would lead to this ion fragment? Is it possible that 42 m/z showing up in both the anthropogenic and background period could be explained by two compounds or types of compounds? Maybe cyanate contributed to the 42 m/z signal during the anthropogenic period and other organic nitrogen during the background period?

We thank the reviewer for this comment. As a result, we have reevaluated the assignment of m/z 42 as C2H4N- and have come to the conclusion that this is likely not the correct molecular assignment for this ion. Based on prior high resolution TDCIMS measurements performed at a variety of sites, we have concluded that the more likely assignment is cyanate ($CNO^-$). This is also based on the likelihood that cyanate would be formed in our negative ionization chemistry, while C2H4N- would be less likely associated with the negative ion chemistry. Our change in assignment of this ion does not affect our current hypothesis that this fragment is associated with nitrogen containing organic species and we stress in the paper that this is a hypothesis, while providing evidence that supports this. While it is possible that the m/z 42 signal could be derived from two or more compounds, they would have to come from similar sources, as the ion fraction and diurnal pattern do not appear to be different between the anthropogenic and background periods.

Based on this reanalysis and response to this reviewer's comment, we have changed the text as follows:

Page 12, lines 291- 303 "The *m/z* 42 fragment observed in this study is not likely anthropogenically derived cyanate since this ion was observed during very clean periods when we expect anthropogenic emissions and biomass burning to be low. In addition, TDCIMS-measured *m/z* 42 during the dry season did not show an increase in ion intensity relative to the wet season (Smith, 2016), which one might expect if this ion were sourced to biomass burning. We hypothesize that this ion is cyanate ($CNO^-$) which we associate with organic nitrogen related to aerosol formation from biogenic emissions of VOCs. Natural emissions of amino acids, water soluble organic species, and other proteinaceous biogenic material have been measured in the gas phase, particle phase and in precipitation across the globe, and have been estimated to account for as much as half or more of the bulk aerosol composition over the Amazon basin (Artaxo et al., 1988, 1990; Kourtchev et al., 2016; Mace et al., 2003; Zhang and Anastasio, 2003). While all prior field measurements in the Amazon basin have been made on particles larger than those measured in this study, similar sources may influence ultrafine particle composition. If true, these observations suggest that organic nitrogen compounds play a crucial role in both ultrafine particle formation as well as growth to large particles, which make this mechanism for particle growth climatologically important in this region."

Page 10 line 294: It is a bit strange the authors used PM2.5 observations of organic nitrogen to justify 42 m/z being organic nitrogen in sub 100 nm particles as they later state that ultrafine particles have unique composition compared to larger particles.

We recognize the wording for this was not right, so we have made the following adjustments:

Page 10, lines 296-303: "We hypothesize that this ion is cyanate ($CNO^-$) which we associate with organic nitrogen related to aerosol formation from biogenic emissions of VOCs. Natural emissions of amino acids, water soluble organic species, and other proteinaceous biogenic material have been measured in the gas phase, particle phase and in precipitation across the globe, and have been estimated to account for as much as half or more of the bulk aerosol composition over the Amazon basin (Artaxo et al., 1988, 1990; Kourtchev et al., 2016; Mace et al., 2003; Zhang and Anastasio, 2003). While all prior field measurements in the Amazon basin have been made on particles larger than those measured in this study, similar sources may influence ultrafine particle composition. If true, these observations suggest that organic nitrogen compounds play a crucial role in both ultrafine particle formation as well as growth to large particles, which make this mechanism for particle growth climatologically important in this region."

Figure 1: It is nearly impossible to read the numbers on the color scale for the particle size distribution color map. Also units of dN/dlogDp should be (cm-3) and not molecules/cm^3. Is there a reason the rainfall scale is wide when the amount of rain does not exceed ~4 mm? Also, the left-hand axes for wind direction and windspeed look like continuations of the observations (i.e. large spike in wind speed and 360 degrees for wind direction). Please make this a bit clearer (and larger so everything is easier to read).

We have updated Figure 1 to be clearer.

Figure 2: Would be helpful to label Manus on this map. Could be a patch of rocks to the unsuspecting reader. Also please add that pink and red traces are from anthropogenic period in the caption.

An updated version of Figure 2 now includes new HYSPLIT back-trajectory frequency plots and do not use different color traces like before. Manaus has been explicitly labeled.

Figure 3: The text in the legend is too small to read. Would be helpful if something was drawn on (a) to indicate when the anthropogenic and background periods were. For (b), consider putting in the m/z for each panel because the reds, greens, blues, and yellows (orange?) look the same.

Figure 3 has been updated with larger font sizes and include the m/z for each diurnal plot. The dates for each time period have been described in the figure caption.

Figure 4: label for the color scale would be useful, especially since there are no -1 correlations (red) but only light oranges that are harder see.

The color scale is explained in the text from lines 343-346. We have also added a label for the color scale indicating what the scale is in reference to.

Figure S1: bisulfate is a clearer marker for anthropogenic influences compared to what?

Rather than noted as "clearer" the sentence now reads:
"Bisulfate (m/z 97) was also chosen for analysis as a marker for anthropogenic influence."

Figure S3: A legend is needed for each panel/color. Is this negative ion? Or positive?

This figure is just example of the ion abundance of negative ions and each plot has been labeled according to the m/z.

Anonymous Referee #2

Summary

The manuscript by Glicker et al. entitled "Chemical composition of ultrafine particles in central Amazonia during the wet season" presents a chemical characterization of ultrafine aerosol particles using a Thermal Desorption Chemical Ionization Mass Spectrometer (TDCIMS) in the course of the GoAmazon field campaign in 2014 and 2015. The authors contrast the ultrafine particle conditions and properties during two different periods, which they call anthropogenic period and biogenic period. They suggest that the chemical results obtained can be regarded as characteristic for the Manaus metropolitan area influence on one hand and clean remote regions further north on the other hand. A certain number of specifically observed ions are discussed in relation to potential sources and processes.

Relevance

This study is clearly an important endeavor since the origin and properties of ultrafine particles in the Amazon are still largely unknown. Particularly, very sparse information is available on the chemical composition of the ultrafine particle fraction. The increasing number and visibility of recent studies on ultrafine Amazonian aerosols underline the relevance of this topic (Wang et al., 2016; Andreae et al., 2018; Fan et al., 2018; Rizzo et al., 2018).

Formal aspects

The paper is well structured and mostly clearly written. The lengths of the text and the number of figures generally seems appropriate although certain crucial aspects may deserve an additional dedicated figure for clarification (see below). The existing body of literature in the field of Amazonian atmospheric research is only partly covered and the study would profit from further literature synthesis.

Scientific assessment

I have a fundamental scientific concern relating to a discrepancy of what the study suggests and concludes vs. what it actually delivers and supports with data. The overarching aim of the work is reflected in the title, announcing a characterization of the "Chemical composition of ultrafine particles in central Amazonia during the wet season". Furthermore, the abstract suggests that the study helps to determine "the chemical species and mechanisms that may be responsible for new particle formation and growth in the region" (p. 1, l. 19-20). Here, the word "region" refers to "central Amazonia" (p. 1, l. 14). In its current state, the study has not convinced me of really being representative for conditions in the central Amazon (as promised by the title, abstract, and sections in the main text), because the plots and data indicated a persistent influence of the Manaus city plume, which seems particularly strong for the ultrafine particles. It seems that the general discrimination into an "anthropogenic period" and a "biogenic period" is oversimplified. I don't doubt that both periods cover distinct conditions and that certain pollutant concentrations were "as much as three times larger" (p.6, l. 200) during the anthropogenic relative to the biogenic period. However, this does not imply that the biogenic period is sufficiently free of anthropogenic pollutants to uncover biogenic processes. In fact, the data/plots did not convince me that the conditions during the "biogenic period" were anywhere close to "clean" (p. 4, l. 131), "natural" (p. 1, l. 15), or even "pristine" (p. 2, l. 69) and "pre-industrial" (p. 1, l. 15) conditions (Hamilton, 2015). The only text section where this is critically reflected can be found in the conclusions: Here the authors state that "influences from anthropogenic sources […] may continuously affect the composition of ultrafine particles observed at the T3 measurement site" (p. 14, l. 388-390). Speaking as the Advocatus Diaboli: 'Could it be that we are looking throughout the entire measurement period at ultrafine particles of anthropogenic origin, only at varying states of dilution and mostly swamping biogenic processes in the background?' If so, the title and argumentation have to be adjusted. This general uncertainty does not harmonize with many statements in the text, such as the very strong conclusion in the last sentence of the study, stating that anthropogenic emissions and processes have a unique role to play in ultrafine particle formation and growth in the Amazon basin".

My recommendation

In principle, the manuscript fits well into ACP in terms of topic, methods, and potential conclusions. However, in the light of my aforementioned concerns, I think that a careful revision and clarification of certain aspects is necessary prior to publication. I am optimistic that addressing the aforementioned points and a careful clarification of what this study specifically provides/means within the wider context of the Amazonian aerosol cycling will ultimately strengthen the points of this work.

Specific major comments

In relation to my aforementioned criticism, some main aspects are outlined in more detail below:

1) Comment relating to terminology: The manuscript frequently and vaguely refers to terms such as "background", "natural", "clean", and "pristine" conditions. It has to be clarified what these terms exactly mean in the context of the presented measurements, the measurement location, and the Amazonian season during which the results were obtained.

Within the context of this paper, the background period differs from the anthropogenic period by the frequency with which air masses reaching the T3 site pass over the City of Manaus. While in our prior draft the concept of "background" was vaguely introduced, the following updates have been made to better describe our definition and how it compares to the anthropogenic period:

Page 3, Line 75-77: To not confuse the reader, the term "background" has been removed from this sentence and edited to the following:

"In the wet season, ambient particle number concentrations often represent pristine, near- natural concentrations and are in the range of 300-600 cm$^{-3}$ (Zhou et al., 2002)."

Page 4, line 130-131: We have edited the brief description of the two air masses sampled in the last paragraph of the introduction:

"Specifically, we focus on ten consecutive days that experienced air masses from both remote, primarily forested regions, as well as from the large metropolitan region of Manaus."

Page 7, beginning line 222: Further insight has been added to the NOAA HYSPLIT back-trajectories to include a more thorough description of the air mass trajectories observed in the background period. This description includes observations of certain times during the background period where there was observed Manaus influence, in additional to more frequent forest influence. More details on our updates to the HYSPLIT model in the following comment.

Page 15, line 407-409: Rather than linking particle composition influence to background sources and processes, the sentence now reads:

"The chemical composition of ultrafine particles in the Amazon basin, as measured during the GoAmazon2014/5, has two distinct influences: sources and processes linked to anthropogenic origin and those related to more natural sources and processes."

2) Comment relating to back trajectories: The back trajectory plot shown in Fig. 2 is (at least) misleading as it shows only a snapshot of the air mass circulation. However, the text and figure caption infer that these snapshot trajectories "show the difference between the types of air masses that travel to the T3 site during the anthropogenic period […] and biogenic period […]". I did a quick trajectory frequency run to visualize conditions over multiple days of the anthropogenic vs. biogenic periods (Fig. R1). Clearly, the trajectory paths of both periods are different: during the anthropogenic period the path is tighter and seems to have more 'contact' to Manaus, whereas during the biogenic period the path is broader and spans over a larger region including Manaus. However, Fig. R1 suggests that Manaus influenced both periods, though to a different extent. Moreover, I don't see evidence that the conditions during the "biogenic period" bring air masses from "clean, remote regions" (p. 4, l. 131). In fact, the trajectories during both periods pass over the Amazon River, which might be quite polluted due to ship traffic, settlements etc. In summary: I think a more systematic trajectory analysis is needed here and a more differentiated discussion of what this means for "biogenic" conditions. Further minor comments in the context: (i) The starting height of the trajectories is not specified. (ii) I could not reproduce the circulation for Mar 23 shown in Fig. 2 suggesting air masses straight from the north. Under which settings was this derived? (iii) In p. 7, l. 202 & 221, it is stated that "[…] during the background period, air masses arrived at the T3 site from the northeast to northwest ~70% of the time (Figure 2)." Where is the number "~70%" coming from? This is not transparently described.

[Figure]

Fig. R1. Back trajectory frequency plots, contrasting "anthropogenic" vs. "biogenic periods".

We thank the reviewer for the suggestion of using HYSPLIT back-trajectory frequencies to give the reader a better insight into the air mass trajectories between the two periods of interest. We have incorporated the reviewer's recommendation for including these trajectories and results are shown below with the correct time periods encompassed, as the reviewer's figure R1 encompassed days that were not in each period. From the back-trajectory frequencies spanning each period, air masses are more frequently traveling from the area of Manaus during the anthropogenic period, while air masses during the background period are travelling more frequently from north of the site. Additionally, we have reframed the description and interpretation of the HYSPLIT model to be more specific and not suggest the background has no Manaus influence. The updated Figure 2 and caption are shown below. In addition, the following edits have been made:

Page 7, starting line 221:

"Wind direction data shown in Figure 1, as well as NOAA HYSPLIT data shown in Figure 2, suggest a reason for the two distinct periods. Back-trajectories show that air masses during the anthropogenic period either pass through Manaus or south of Manaus prior to arrival at the T3 site. During this period, air masses most frequently passed over the main roadway that connects Manaus with Manacapuru, a neighboring city with a population of 93,000. Along this roadside are homes, agriculture and brick kilns, all of which contribute to local gas and particle emissions. In contrast, during the background period, air masses arrived at the T3 site most frequently from the north east and west. Air masses that were measured at the site typically originated from densely forested regions northeast to west of Manaus. Less frequent were periods where air masses reaching the site originated from east and were influenced by the Manaus metropolitan area. For example, during the evening of 21 March there was a period of increased number concentration and, as winds were quite stagnant at night, it is possible that a local emission source could have impacted the site during that period. Wind direction on this day corresponded with air masses arriving to the T3 site from the Manaus area."

3) Comment relating to the 'overview' Figure 1: Figure 1 seems to be meant as an overview plot to

[Figure]

**Figure 2:** Back trajectory frequencies performed using HYSPLIT, showing the different air masses that travel to the T3 site during the anthropogenic period and background period. For each period, twenty trajectories were used to determine integrated frequencies spanning the five days of each period (14 Mar-19 Mar for the anthropogenic and 20 Mar-25 Mar for the background period). Each trajectory duration was for 72 hours. The color scale indicates the frequency of which air masses pass over that area, with the warmer color being more frequently passed over.

illustrate the overall conditions during the measurement period. It puts a major focus on local meteorological parameter, which is of course helpful. However, it only provides very sparse aerosol context given that a broad range of aerosol data was measured during the GoAmazon campaign. Particularly, I feel that some basic time series such as on total particle concentration and black carbon (BC) concentration would be very helpful to illustrate the contrast between both periods. In particular, BC could help to identify periods without detectable anthropogenic aerosol influence. Moreover, a particle concentrations in the ultra-fine particle size range (i.e., <30 nm) would be very interesting/relevant as this is the focus of the whole study. Maybe also particle concentrations for the Aitken and accumulation mode ranges would be helpful, since the typical multimodal shape of the Amazonian aerosol distribution (Artaxo et al., 2013; Andreae et al., 2015) can hardly be seen in the SMPS contour plot in Fig. 1.

Per the suggestions of both this reviewer and Reviewer 1, additional analysis of black carbon mass concentration data was added to explore the potential impact of biomass burning on ultrafine particle composition. As a result, an additional supplemental figure has been added (Fig. S4) and the following text was added in Section 3.2:

Page 12, lines 322-333: "One period of elevated potassium ion ratio was observed at the end of the day on 22 March. To differentiate between potential sources of potassium in these ultrafine particles, whether it be of primary biological or biomass burning influence, mass concentrations of black carbon during this ten-day period of interest were used to examine the extent of influence of biomass burning on the presence of potassium (Fig. S5). During the anthropogenic period, with significantly elevated concentrations of black carbon, minimal potassium fraction was measured. At times of low black carbon mass concentrations during the background period, like on 20 March, there was some fraction of potassium observed. During the period of highest fraction observed on the night of 22 March, there were slightly elevated mass concentrations of black carbon. While partially elevated black carbon mass concentrations on 22 March may be connected to the large potassium ion fraction, at times with even more significant biomass burning influence, there was minimal potassium. The larger fraction of potassium observed during the background period, as opposed to the anthropogenic period, may be connected to potassium rich biological particles or the rupturing of biological spores (China et al., 2016; Pöhlker et al., 2012)."

We have also added the total number concentration for sub 100 nm particles to Figure 1. Additional particle size distribution measurements and information have been included as updates to the paper and are addressed in the next comment.

4) Comment on aerosol size distributions: The focus of this study is the analysis of ultrafine particles, which generally have a rather low abundance in the Amazon. Actually, the low abundance (though not absence) is a main reason why ultrafine particle studies in the Amazon are so exciting/relevant. For illustration, I compiled some size distribution plots from previous studies in Fig. R2 – all of them show the characteristic multimodal shape without a clearly resolved nucleation mode. How does this relate to Fig. 1 in the present study? How representative are the size distributions in Fig. 1 for the Amazon region? All episodes with the very high abundance of ultrafine particles in Fig. 1 (also during the "biogenic period") differ substantially from the distributions in Fig. R2, suggesting an impact of Manaus or even more local pollution. What about the 'as clean as it gets' conditions in Fig. 1 – do they resemble the plots in Fig. R2? What is missing in the text is a dedicated comparison of the observed size distributions in Fig. 1 with the existing literature (e.g., those in Fig. R2). Ideally, the authors could add a plot with dN/dlogD distributions that directly compares the condition during their measurement period(s) with characteristic distributions from previous publications. **In summary:** The study aims at "determining the chemical species and mechanisms that may be responsible for new particle formation and growth in the region" (p. 1, l. 19/20). I think that it is not convincingly shown (yet) that the aerosol size distributions, which underlie the TDCIMS analysis, resemble the previously published size distributions that are typical for the Amazon region. It could well be that certain episodes in the measurement period resemble those characteristic conditions, however, it is very hard to see from Fig. 1. Putting the conditions of the present study into a broader literature context will likely strengthen the case of this work substantially. Further minor comment in the context: The color scale in Fig. 1 is confusing since most of the shown concentration range is red. What is the purpose for doing it like that?

[Figure]

Fig. R2. Previously reported aerosol size distributions from the Amazon region (Gunthe et al., 2009; Artaxo et al., 2013; Andreae et al., 2015; Pohlker et al., 2016; Rizzo et al., 2018).

With regards to the particle size distributions measured during the GoAmazon campaign at the T3 site, these ten days are representative of measurements during the whole wet season IOP. The figure below is the campaign SMPS measurements for March of IOP1.

[Figure]

Additionally, below are size distributions of particles measured on the morning (between midnight at 9:00) on 16 March and 21 March, representing a series of distributions from the anthropogenic and background periods respectively.

[Figure]

While during our anthropogenic period, there aren't clear signs of a bimodal distribution, there is one mode of increased particle number concentration, peaking at around 50 nm. Rizzo et al. (2018) average size distribution from the dry season also featured this unimodal distribution. Specifically looking at the size distribution for the morning of 21 March, there is a clearer bimodal distribution, with one peak of on average 1000 cm$^{-3}$ at 50 nm and 500 cm$^{-3}$ at 150nm. This bimodal distribution seen during the morning of 21 March is at comparable sizes to the reviewer's distributions supplied for Artaxo et al. 2013, Rizzo et al. 2018, Gunthe et al. 2009 and Pohlker et al. 2016. These figures above have been added to the supplemental as Fig. S2 and the following has been added to the paper:

Page 7, lines 207-210: "Particle size distributions for the background period were comparable to previous measurements in the Amazon basin, featuring a bimodal distribution with peaks at roughly 50 nm and 150 nm and peak concentrations of approximately 10$^3$ particles cm$^{-3}$ (Fig S2) (Artaxo et al., 2013; Gunthe et al., 2009; Pöhlker et al., 2016; Rizzo et al., 2018)."

Anonymous Referee #3

General Comments:

This manuscript reports the composition of ultrafine particles during the wet season in central Amazonia as measured by a Thermal Desorption Chemical Ionization Mass Spectrometer (TD-CIMS). The top five abundant ions by signal from each of negative and positive ion modes are reported for a ten-day period representing anthropogenically-influenced and background conditions. The authors find that particulate bisulfate is elevated during the anthropogenic period, though omnipresent, and that organic nitrogen is characteristic of background airmasses. 3-methylfuran (ascribed to IEPOX chemistry) is the dominant component in positive ion mode and interpreted to contribute to new particle growth and formation processes. Finally, the authors find using principal component analysis that ultrafine particle composition can be divided into two clusters, one mostly comprised of organics, and the other comprised of inorganic ions, both distinct from a third cluster with most AMS PM1 measured constituents, indicating unique sources/chemistry for ultrafine and PM1 particles. Overall, this work provides novel measurement of ultrafine particle composition in central Amazonia and would be appropriate for publication in ACP after the following comments are addressed. It is generally written clearly, but lacks some depth in providing additional insight from the measurements. For example, the discussion on PCA analysis could provide more insight into the observed correlations between species/clusters, and as written tends to just reiterate earlier descriptions of the ascribed sources for TDCIMS ion assignments.

Specific Comments:

1) Line 60: In addition to Alves et al., 2016, consider adding citation to the following:

a. Jardine, K. J., Yañez Serrano, a., Arneth, a., Abrell, L., Jardine, A. B., Van Haren, J., Artaxo, P., Rizzo, L. V., Ishida, F. Y., Karl, T., Kesselmeier, J., Saleska, S. and Huxman, T.: Withincanopy sesquiterpene ozonolysis in Amazonia, J. Geophys. Res. Atmos., 116(19), 1–10, doi:10.1029/2011JD016243, 2011.

b. Jardine, A. B., Jardine, K. J., Fuentes, J. D., Martin, S. T., Martins, G., Durgante, F., Carneiro, V., Higuchi, N., Manzi, A. O. and Chambers, J. Q.: Highly reactive lightdependent monoterpenes in the Amazon, Geophys. Res. Lett., 42(5), 1576–1583, doi:10.1002/2014GL062573, 2015.

c. Shrivastava, M. K., Andreae, M. O., Artaxo, P., Barbosa, H. M. J., Berg, L. K., Brito, J., Ching, J., Easter, R. C., Fan, J., Fast, J. D., Feng, Z., Fuentes, J. D., Glasius, M., Goldstein, A. H., Alves, E. G., Gomes, H., Gu, D., Guenther, A., Jathar, S. H., Kim, S., Liu, Y., Lou, S., Martin, S. T., McNeill, V. F., Medeiros, A., de Sá, S. S., Shilling, J. E., Springston, S. R., Souza, R. A. F., Thornton, J. A., Isaacman-VanWertz, G., Yee, L. D., Ynoue, R., Zaveri, R. A., Zelenyuk, A. and Zhao, C.: Urban pollution greatly enhances formation of natural aerosols over the Amazon rainforest, Nat. Commun., 10(1), 1046, doi:10.1038/s41467-019-08909-4, 2019.

d. Yáñez-Serrano, A. M., Nölscher, A. C., Williams, J., Wolff, S., Alves, E. G., Martins, G. A., Bourtsoukidis, E., Brito, J. F., Jardine, K. J., Artaxo, P. and Kesselmeier, J.: Diel and seasonal changes of biogenic volatile organic compounds within and above an Amazonian rainforest, Atmos. Chem. Phys, 15, 3359–3378, doi:10.5194/acp-15-3359-2015, 2015.

e. Yee, L. D., Isaacman-Vanwertz, G., Wernis, R. A., Meng, M., Rivera, V., Kreisberg, N. M., Hering, S. V, Bering, M. S., Glasius, M., Upshur, M. A., Bé, A. G., Thomson, R. J., Geiger, F. M., Offenberg, J. H., Lewandowski, M., Kourtchev, I., Kalberer, M., de Sá, S. S., Martin, S. T., Alexander, M. L., Palm, B. B., Hu, W., Campuzano-Jost, P., Day, D. A., Jimenez, J. L., Liu, Y. J., Mckinney, K. A., Artaxo, P., Viegas, J., Manzi, A., Oliveira, M. B., De Souza, R., Machado, L. A. T., Longo, K. and Goldstein, A. H.: Observations of sesquiterpenes and their oxidation products in central Amazonia during the wet and dry seasons, Atmos.

Chem. Phys, 18, 10433–10457, doi:10.5194/acp-18-10433-2018, 2018.

In addition to Alves et al., 2016, the following citations have been added:
Jardine et al., 2011 and 2015; Yanez-Serrano et al., 2015 and Yee et al., 2018. We have decided to not include the Shrivastava et al. paper as this is for modeling work, but have included this work within proper context in the introduction as well.

2) Lines 73-78: It might be worthwhile to define "ultrafine", "Aitken", "accumulation", and "coarse mode" particles for readers less familiar with these distinctions in Dp ranges.

     Sentence has been changed to the following:
Page 3, Lines 73-75: "During the wet season (December through March), the region is dominated by natural emissions, as accumulation mode (particle diameters between 0.1 and 2.5 µm) and coarse mode (diameters above 2.5 µm)  particles tend to be lower in concentration due to wet deposition (Andreae, 2009)."

3) Line 101: Please rephrase "…can have an oversized impact…" as it is not very scientifically clear wording.

     Sentence has been changed to the following:
Page 3, Lines 101-103: "Once formed, ultrafine particles can be key participants in a variety of atmospheric processes. One example of this is the subject of a recent study by Fan et al. (2018) has suggested that ultrafine particles can increase the convective intensity of deep convective clouds."

4) Line 156: Can the authors also include the MS for positive ion mode? Why was m/z 75 not selected for regular measurement considering its ion intensity is relatively large?

     We have now included an example of a background subtracted for positive ion mode as well (Fig S1). These two are just examples of the full mass spectra observed. During the campaign itself, ions were identified based on their abundance in various spectra and selected for continued measurement.

5) Line 160: Please provide additional information in Smith, 2016 under references to make it easier to find.

     The updated reference is as follows:
Smith, J. N.: Thermal Desorption Chemical Ionization Mass Spectrometry during GoAmazon2014/5, data portal: https//iop.archive.arm.gov/arm-iop/2014/mao/goamazon/T3/smith-tdcims, 2016

6) Line 161: Can you specify the threshold for "low concentrations" of ultrafine particles?

We acknowledge the term low concentrations is not very descriptive, what we meant to say that lower concentrations of ultrafine particles were observed in the dry season compared to the wet season. The following sentence has therefore been changed:
Page 5, line 161-162: "IOP2 was characterized by comparatively lower concentrations of ultrafine particles, which is consistent with prior observations (Martin et al., 2010; Rizzo et al., 2018)."

7) Figure 1: For size distribution plot, why are the units of intensity for dN/dlogdP in molec/cm^3 rather than #/cm^3 considering that particle concentrations have been discussed earlier in manuscript as #/cm^3?

     Per the suggestions of both this reviewer and Reviewer 1, the labeling for the size distribution plot has been corrected.

8) Lines 224-238: This is very interesting analysis. Would the authors be able to infer from this an average % increase in loading on top of "background" conditions that is attributable to anthropogenic influence, assuming that the "background" composition from the March 15-March 19 period is approximately same for the March 20-25 period?

The following information has been added and Fig. S3 (included below) has been updated as such:
Page 8, line 240-242: "Assuming that the background contribution to the mass of particles remains constant between each time period, the average mass loading of ultrafine particles increased by a factor of 3 due to anthropogenic influence (Fig. S3)."

[Figure]

**Fig. S3:** Diurnal patterns of the estimated mass collected on the TDCIMS Pt filament during collection. The blue crosses are average values, the boxes show 25th and 75th percentiles as well as medians, and the whiskers show maximum and minimum values. a) Anthropogenic period: in which no regular diurnal pattern is observed. b) Background period: characterized by peaks in collected mass in mid-afternoon and at least half the mass collected compared to the anthropogenic period. The horizontal dashed lines represent the average mass collected for each period, with the average mass collected for anthropogenic period being 126 ± 124 ng and for the background period being 39.9 ± 41.2 ng.

9) Figure S3: Please include figure legends for the ions shown in these diurnal profiles and specify that this is negative ion mode in caption.

Labels of the ions have been added to Figure S3, as well as noting these are negative ions.

10) Lines 264-269: Can the authors clarify if bisulfate ion as indicator of particulate sulfate can also include natural/background sources of sulfate? Since it has been previously established that there are a lot of natural sources of sulfate (e.g. DMS) (Andreae et al., 1990; Andreae and Andreae, 1988) as well as background levels (long-range transport including anthropogenic) (de Sá et al., 2017), would the authors anticipate these sources to be contributing to the majority of the bisulfate anion signal during 19 Mar to 26 Mar 2014?

Comment #14 applied to this comment and sentence has been moved.

11) Line 293: Please specify basis of 55-95% of PM as mass basis, etc.

In response to this and reviewer 1, the following sentence has been changed:
Page 11, lines 296-299: "Natural emissions of amino acids, water soluble organic species, and other proteinaceous biogenic material have been measured in the gas phase, particle phase and in precipitation across the globe, and have been estimated to account for as much as half or more of the bulk aerosol composition over the Amazon basin (Artaxo et al., 1988, 1990; Kourtchev et al., 2016; Mace et al., 2003; Zhang and Anastasio, 2003)."

12) Lines 305-306: Based on the diurnal profile of m/z 83 assigned as 3-methylfuran, the authors could better support the claim for IEPOX as a proposed source by comparing with diurnal profiles of gas-phase isoprene oxidation products by PTR-MS (Liu et al., 2016, 2018), particle phase isoprene oxidation products by SV-TAG Figure 1c, d (Isaacman-VanWertz et al., 2016), and AMS IEPOX-SOA PMF factor Figure 4b (de Sá et al., 2018) ? Does it make sense for Isoprene + OH ☐ IEPOX to occur 8:00-10:00 UTC and peak, followed by minimum 14:00-16:00 UTC, and then build again?

   Similar results in diel variability (or lack thereof) were observed in SV-TAG measurements presented in Figure 1c,d (shown below). There was a significantly weaker diurnal variability observed for the particle phase IEPOX derived species compared to the stronger afternoon peak of gas phase species.

[Figure]

(Figure 1, Isaacman-VanWertz, 2016)
In the previous version of this paper, the emphasis on the diurnal variability and peak at 8:00-10:00, was a bit too strong of a statement considering the actual data shown in Figure 3. The variability for this species, on average, fluctuated between 40-70% of the total fraction observed, but with larger error bars than other measured species. This is indicative that there is no clearly strong diurnal pattern, similar to work shown in figure above. The follow addition and edits have been made to this section and are below:

Page 12, starting line 307: "Airborne observations in the Amazon suggest that isoprene SOA can be formed in the boundary layer under certain conditions, which is confirmed by these observations (Allan et al., 2014). Since this ion is a marker of isoprene epoxydiol (IEPOX) species present in the particle phase, this confirms a role for isoprene and isoprene derivatives in the growth of ultrafine particles. Little variability in the diel pattern for m/z 83 is observed, similar to other particle phase measurements of IEPOX derivatives reported for the GoAmazon2014/5 campaign by Isaacman-Vanwertz, et al. (2016). In that study, weak diurnal patterns for particle phase isoprene oxidation products were also observed, even while gas phase concentrations of these species increased in the afternoon."

13) Figure 4: Can the chemical assignments be added after the TDCIMS measured m/z's for ease of chemical interpretation just looking at figure, (e.g. m/z 89 hydrogen oxalate, m/z 59 acetate, etc.)

   To not look cramped and take up too much space, chemical assignments have been added to the figure caption. The figure caption for Figure 4 now reads as follows:

**Figure 4**: Principal Component Analysis (PCA) of TDCIMS and AMS data. Refer to text for details on the interpretation of these plots. PCA results in which species are grouped into hierarchical clusters, with clusters denoted within weighted black lines. Species are ordered by decreasing correlation to the first principal component from the top to bottom. TDCIMS chemical assignments for fragments are m/z 89 (hydrogen oxalate), m/z 59 (acetate), m/z 42 (cyanate), m/z 60 (trimethyl ammonium), m/z 83 (3-methylfuran), m/z 36 (ammonium hydrate), m/z 97 (bisulfate), m/z 35 (chloride) and m/z 39 (potassium).

14) Lines 371-373: Move this explanation of natural bisulfate sources up in manuscript based on Specific Comment regarding Lines 264-269) above.

This explanation of natural sources of sulfate has been moved to lines 274-277. The following sentence has been added to the last paragraph describing the bisulfate significance in the PCA analysis: Page 15, Line 391-393: "However, in-basin emissions of sulfate gaseous precursors, like dimethyl sulfide and hydrogen sulfide, could be linked to the measured bisulfate fraction during the entire ten-day period with anthropogenic sources of sulfate increasing this background level during the anthropogenic period."

15) Section 3.3. Authors should include more analysis and interpretation of Figure 3. Can any of these questions below be answered with the PCA analysis:

a. What do the authors make of the fact that AMS chloride and TDCIMS m/z 35 chloride are in the same cluster despite different size distribution ranges of the two measurement techniques?

Figure 3 shows that AMS chloride is unique among other measured AMS species in that it is only weakly correlated to each of these other AMS-derived species. This fact, as well as a similarly weak correlation with the other TDCIMS-derived ions in Cluster 3, are the likely reasons for its assignment to Cluster 3.
The following addition has been added to the paper:
Page 15, lines 395-399: "The clustering of AMS chloride with TDCIMS species in Cluster 3 might suggest similar sources of chloride in both ultrafine particles and PM2.5. However, it is worth noting that AMS chloride also very weakly correlated with the other species measured by the AMS. For this reason, its inclusion in this cluster indicates both that AMS chloride is similar to TDCIMS-derived Cluster 3 species and different enough so as not to cluster with the other AMS species."

b. Why does TDCIMS measured m/z 42 (organic nitrogen) negatively correlate with AMS nitrate? What implications does this have in terms of sources of organic nitrogen between ultrafine and PM1?

AMS Nitrate has been associated in de Sa et al. (2018, 2019) to both organic and inorganic forms of nitrate. Of the already small AMS concentrations of total nitrate measured, the ratio for organic nitrates was assumed to be a factor of 2.25 lower than that of inorganic nitrates (Supplemental, de Sa et al., 2018). The slightly negative correlation between our organic nitrogen and AMS nitrate could be due to the fact that the latter can be attributed more to inorganic nitrates. Additional insight has been added to the paper.

Page 14, lines 379-383 "Additionally, TDCIMS-measured cyanate (*m/z* 42) are weakly and negatively correlated to AMS-measured nitrate. During the anthropogenic period (14 March through mid-morning 19 March), higher levels of inorganic nitrate were observed by AMS compared to the organic form (de Sá et al., 2018). This higher mass concentration of nitrate attributed to inorganic nitrate, as opposed to organic nitrate which would be more similar to TDCIMS-measured cyanate, should explain the slight negative correlation between the two."

c. Lines 375-379: Why was AMS-measured K+ not included in the PCA analysis to see if it is distinct/similar to TDCIMS across size distributions?

Potassium was not included in AMS measurements during the campaign. All species measured were included in the PCA analysis.

16) Lines 379-381: Repetitive with lines 365-367.

Technical Corrections:
1) Line 56: Delete "is."
2) Line 124: Change "process" to "processes."
3) Lines 131: Change "mass" to "masses."
4) Lines 174-181: Reorder references to ARM, 2018a-d so they appear in alpha order.
5) Line 353: Delete "and" at start of line.
6) Line 395: No need to capitalize "Area."
7) Line 396: Change "underscore" to "underscores"

Every technical correction above has been changed.

[revised manuscript text omitted]
 $\text{C}_2\text{H}_4\text{N}$CNO$^-$ (organic nitrogen speciescyanate, $m/z$ 42), $\text{C}_2\text{H}_3\text{O}_2^-$ (acetate, $m/z$ 59), $\text{HSO}_4^-$

(bisulfate, $m/z$ 97), Cl$^-$ (chloride, isotopes $m/z$ 35 and 37) and $\text{HC}_2\text{O}_4^-$ (hydrogen oxalate, $m/z$ 89). The six most abundant positive ions measured were attributed to $\text{NH}_4^+(\text{H}_2\text{O})$ (ammonium hydrate, $m/z$ 36), K$^+$ (potassium, isotopes

$m/z$ 39 and 41), $\text{C}_3\text{H}_{10}\text{N}^+$ (trimethyl ammonium, $m/z$ 60), $\text{C}_5\text{H}_7\text{O}^+$ (protonated 3-methylfuran, $m/z$ 83), $\text{C}_5\text{H}_8\text{NO}^+$ ($m/z$

98), and $\text{C}_7\text{H}_9\text{O}_2^+$ ($m/z$ 125). We will refer to $\text{C}_5\text{H}_8\text{NO}^+$ ($m/z$ 98), and $\text{C}_7\text{H}_9\text{O}_2^+$ ($
[revised manuscript text omitted]